# Role of Gut Microbiome in Oncogenesis and Oncotherapies

**DOI:** 10.3390/cancers18010099

**Published:** 2025-12-29

**Authors:** Renuka Sri Sai Peddireddi, Sai Kiran Kuchana, Rohith Kode, Saketh Khammammettu, Aishwarya Koppanatham, Supriya Mattigiri, Harshavardhan Gobburi, Suresh K. Alahari

**Affiliations:** 1Mansfield Kaseman Health Clinic, Rockville, MD 20850, USA; 16770700@igmcri.edu.in; 2Department of Internal Medicine, Kakatiya Medical College, Warangal 506007, India; saikirankuchana03@gmail.com (S.K.K.); koderohith4@gmail.com (R.K.); 3Apollo Institute of Medical Sciences and Research, Hyderabad 500096, India; saketh0803@gmail.com; 4Andhra Medical College, Visakhapatnam 530002, India; aishwaryakoppanatham@gmail.com; 5Katuri Medical College and Hospital, Guntur 522019, India; supriyamattigiri@gmail.com; 6Department of Internal Medicine, Osmania Medical College, Hyderabad 500095, India; vharsha811@gmail.com; 7Department of Biochemistry and Molecular Biology, Louisiana State University Health Sciences Center at New Orleans, New Orleans, LA 70112, USA

**Keywords:** gut microbiome, dysbiosis, oncogenesis, oncotherapy, cancer immunotherapy, probiotics, fecal microbiota transplantation (FMT)

## Abstract

The gut microbiome, which consists of trillions of microorganisms living in the human digestive tract, plays an important role in regulating immunity, metabolism, and inflammation. Growing evidence suggests that disruptions in this microbial community may influence how cancers develop and how patients respond to cancer treatments. In this review, we summarize current research linking the gut microbiome to thirteen common cancers, including cancers of the gastrointestinal tract and cancers affecting other organs such as the breast, lung, brain, and skin. We highlight how certain microbes may promote cancer-related inflammation or DNA damage, while others appear to support immune responses that improve the effectiveness of cancer therapies, particularly immunotherapy. We also discuss emerging strategies aimed at modifying the microbiome, such as diet, probiotics, and fecal microbiota transplantation, and their potential role in future cancer care. Overall, this review provides an accessible overview of how the gut microbiome may influence cancer development and treatment.

## 1. Introduction

The human gut microbiome, comprising trillions of microorganisms including bacteria, viruses, fungi, and archaea, functions as a “virtual organ” that interacts dynamically with host metabolism, immunity, and overall health (Figure 1) [1]. These microbial communities inhabit the gastrointestinal (GI) tract in a finely balanced state, contributing to nutrient metabolism, epithelial barrier integrity, immune regulation, and protection against pathogens [2]. A balanced microbiome is thus essential for homeostasis, while disruption of this balance is referred to as dysbiosis and is increasingly associated with diverse pathological processes including cancer [3,4].

Protective microbes such as *Akkermansia*, *Ruminococcus*, and *Faecalibacterium* are linked to favorable immune modulation, while pro-tumor taxa such as *Fusobacterium*, *Porphyromonas*, *Helicobacter pylori*, and *Streptococcus* are associated with oncogenic inflammation, immune suppression, and tumor progression across multiple organ systems [5].

Dysbiosis typically involves loss of beneficial commensals, overgrowth of pathogenic species, or reduced microbial diversity [6]. Its systemic effects are mediated through multiple pathways: (i) chronic low-grade inflammation caused by microbial products such as lipopolysaccharides (LPSs) and flagellins that activate host Toll-like receptors (TLRs); (ii) direct genotoxicity from bacterial metabolites such as colibactin or nitrosating agents; (iii) disruption of bile acid metabolism leading to accumulation of carcinogenic secondary bile acids; and (iv) impairment of the gut barrier, which allows translocation of microbes and inflammatory mediators into systemic circulation [7]. These processes collectively foster a microenvironment conducive to carcinogenesis by promoting DNA damage, aberrant cell proliferation, and immune evasion.

The significance of the gut microbiome in oncology is twofold; first, as a contributor to oncogenesis, and second, as a determinant of oncotherapy outcomes. Oncogenesis is influenced by microbial communities that either directly induce DNA mutations or create chronic inflammatory states. For example, *Fusobacterium nucleatum* has been implicated in colorectal cancer (CRC) progression by binding to epithelial cells via its adhesin FadA, activating Wnt/β-catenin signaling, and dampening natural killer (NK) cell activity [8]. Similarly, *Helicobacter pylori* infection is a well-established cause of gastric adenocarcinoma through chronic gastritis, inflammatory cytokine release, and DNA damage mediated by virulence factors CagA and VacA [9]. In hepatocellular carcinoma, alterations in bile acid metabolism mediated by gut dysbiosis disrupt the FXR–TGR5 signaling axis, contributing to tumor growth [10]. Across these and other cancers, dysbiosis not only initiates tumorigenesis but also shapes progression and prognosis.

Equally important is the role of the microbiome in modulating oncotherapy. Clinical and preclinical studies increasingly demonstrate that gut microbes influence the efficacy and toxicity of chemotherapy, radiotherapy, and particularly immunotherapy. For instance, response to immune checkpoint inhibitors (ICIs) such as anti-Programmed cell death protein 1 (PD-1) and anti-Cytotoxic T-lymphocyte-associated protein 4 (CTLA-4) therapies (Figure 2) are strongly linked to gut microbial diversity and the presence of taxa such as *Akkermansia muciniphila* and *Bifidobacterium longum* [11,12]. Fecal microbiota transplantation (FMT) from ICI responders into non-responders has been shown to restore sensitivity to treatment, underlining the therapeutic potential of microbiome modulation [13]. Furthermore, dietary fiber intake and probiotic supplementation have been associated with enhanced antitumor immunity and reduced treatment-related toxicities [14]. These findings highlight the microbiome as a risk factor and a therapeutic ally in cancer management.

Despite rapidly growing evidence, most prior reviews have focused on individual cancers such as CRC or gastric cancer, where the microbiome cancer connection is strongest. However, the gut microbiome exerts systemic effects that transcend organ boundaries, influencing malignancies within and outside the GI tract. For example, dysbiosis has been linked not only to GI cancers but also to oral squamous cell carcinoma, cervical cancer, prostate cancer, and even melanoma, where microbiome composition predicts response to immunotherapy. Thus, a broad-spectrum review integrating evidence across multiple cancers (Figure 1) is timely and necessary. Such an approach allows recognition of common mechanisms (inflammation, genotoxicity, barrier dysfunction, metabolic alterations) while highlighting cancer-specific microbial drivers and therapeutic implications.

This review aims to synthesize current evidence on the role of the gut microbiome in both oncogenesis and oncotherapies, with a focus on thirteen cancers in which microbiome–cancer associations are most consistently reported and clinically relevant: colorectal cancer (CRC), gastric cancer, hepatocellular carcinoma, gallbladder cancer (GBC), esophageal cancer, pancreatic cancer, oral squamous cell carcinoma, cervical cancer, prostate cancer, brain cancer, breast cancer, lung cancer (LC), and melanoma [17]. By integrating mechanistic insights and therapeutic perspectives across these malignancies, we seek to provide a broad conceptual framework for understanding the microbiome–cancer axis and to highlight potential avenues for translating microbiome-related findings into clinical oncology.

### 1.1. Literature Selection and Narrative Synthesis

This article is designed as a narrative review synthesizing current mechanistic and translational evidence on the role of the gut microbiome in oncogenesis and oncotherapies. Relevant literature was identified through targeted searches of PubMed and Web of Science, focusing primarily on studies published between 2004 and 2025. Search terms included combinations of gut microbiome, dysbiosis, cancer, oncogenesis, immunotherapy, chemotherapy, and fecal microbiota transplantation.

Priority was given to peer-reviewed studies providing mechanistic insights, clinical associations, or interventional evidence linking microbiome alterations to cancer initiation, progression, or therapeutic response. Given the heterogeneity of study designs and outcomes across cancer types, evidence was synthesized qualitatively rather than through formal meta-analysis. No formal risk-of-bias assessment was performed, consistent with the narrative nature of this review.

### 1.2. Review of Oncogenesis and Oncotherapies

The gut microbiome influences cancer through chronic inflammation, immune modulation, metabolite production, and direct genotoxic effects [18]. However, its impact is not uniform across all malignancies. For this review, thirteen cancers were selected where microbiome cancer associations are best supported by mechanistic and clinical evidence. These include six GI cancers with direct gut–tumor interactions and seven extra-GI cancers with systemic or local microbiome–immune influences.

## 2. Gastrointestinal Cancers

### 2.1. Colorectal Cancer

CRC is the most extensively studied malignancy in relation to the microbiome. *Fusobacterium nucleatum*, enterotoxigenic *Bacteroides fragilis*, and colibactin-producing *Escherichia coli* are enriched in tumors, where they promote DNA damage, activate Wnt/β-catenin signaling, and suppress immune surveillance [19]. Dysbiosis also predicts resistance to chemotherapy (e.g., 5-Fluorouracil (5-FU)) and ICIs [20]. CRC was chosen as the most extensively studied cancer with well-characterized microbiome associations and strong mechanistic and therapeutic evidence [21,22,23,24,25].

#### 2.1.1. Microbiome in Oncogenesis

CRC is the most well-established cancer linked to microbiome dysbiosis. *Fusobacterium nucleatum* promotes carcinogenesis by binding E-cadherin through its FadA adhesin, activating β-catenin signaling, and recruiting tumor-associated myeloid cells that suppress antitumor immunity [21,22]. Enterotoxigenic *Bacteroides fragilis* secretes fragilysin, a metalloproteinase that cleaves E-cadherin, increasing permeability and inflammation [23]. Colibactin-producing *Escherichia coli* directly induces DNA double-strand breaks, fueling genomic instability [24,25]. Reduced abundance of butyrate-producing taxa such as *Faecalibacterium prausnitzii* diminishes anti-inflammatory metabolites, exacerbating tumor-promoting inflammation [26].

#### 2.1.2. Microbiome in Therapy Response

Gut microbiota have an effect on therapy outcomes in CRC. *F. nucleatum* confers resistance to 5-FU and oxaliplatin by activating autophagy and ferroptosis-related pathways [27,28]. Conversely, butyrate-producing bacteria enhance epithelial apoptosis and improve chemotherapy efficacy [26]. Immunotherapy response is linked to higher microbial diversity and enrichment of *Akkermansia muciniphila* and *Bifidobacterium longum* [29]. FMT from immunotherapy responders has been shown in murine models to restore anti-PD-1 efficacy in non-responders [30]. These findings underscore microbiome-targeted approaches as adjuncts in CRC therapy.

### 2.2. Gastric Cancer

*H. pylori* infection remains one of the clearest examples of microbe-induced carcinogenesis with Cytotoxin-associated gene A (CagA) and Vacuolating cytotoxin A (VacA) virulence factors driving chronic gastritis, epithelial transformation, and DNA damage [31]. Beyond *H. pylori*, dysbiosis with increased *Prevotella* and nitrosating bacteria exacerbates inflammation and mucosal injury [32]. These features make gastric cancer central to understanding how gut microbial communities initiate and sustain tumorigenesis [33].

#### 2.2.1. Microbiome in Oncogenesis

*H. pylori* is the archetypal microbial carcinogen, recognized by World Health Organization (WHO) as a class I carcinogen [34]. Its virulence factors CagA and VacA induce DNA damage, activate Mitogen-activated protein kinase (MAPK) and NF-κB signaling, and create a pro-inflammatory gastric environment [31]. Beyond *H. pylori*, gastric dysbiosis involves overrepresentation of nitrosating bacteria (*Neisseria*, *Haemophilus*) that generate carcinogenic N-nitroso compounds [32]. Loss of *Lactobacillus* and *Bifidobacterium* further impairs mucosal defense [35]. The combined effect is chronic gastritis progressing to atrophy, metaplasia, and adenocarcinoma [36].

#### 2.2.2. Microbiome in Therapy Response

The gastric microbiome also modulates therapy. *H. pylori* infection decreases efficacy of standard chemotherapy by altering p53 signaling [37]. Dysbiosis affects immune checkpoint blockade (ICB) response, with *Clostridiales* enrichment associated with better outcomes [38]. Probiotics (*Lactobacillus rhamnosus GG*) can reduce side effects of chemotherapy and may enhance mucosal healing post-eradication therapy [39]. Thus, microbiome modulation could both prevent and improve therapeutic outcomes in gastric cancer.

### 2.3. Hepatocellular Carcinoma

The gut–liver axis highlights how microbial products and metabolites affect hepatic oncogenesis. Dysbiosis alters bile acid metabolism, impairing Farnesoid X receptor/Takeda G protein-coupled receptor 5 (FXR/TGR5) signaling and promoting carcinogenesis (Figure 3) [40]. Translocation of oral-origin bacteria (*Veillonella*, *Streptococcus*) into cirrhotic guts further aggravates inflammation [41]. Importantly, microbiome composition predicts response to immunotherapy, with *Akkermansia muciniphila* and *Ruminococcaceae* associated with favorable outcomes [42]. Thus, HCC exemplifies metabolic and immune interactions mediated by gut microbes.

#### 2.3.1. Microbiome in Oncogenesis

The gut–liver axis explains the critical role of the microbiome in HCC. Dysbiosis reduces bile salt hydrolase-producing taxa, resulting in an accumulation of carcinogenic secondary bile acids [40]. Translocation of bacterial endotoxins (LPS) across a compromised gut barrier activates TLR4 on hepatocytes and Kupffer cells, inducing chronic inflammation and fibrosis [44]. Overgrowth of oral-origin bacteria (Veillonella, Streptococcus) in cirrhotic livers further aggravates hepatocarcinogenesis [41,45].

#### 2.3.2. Microbiome in Therapy Response

Microbiome also predicts response to systemic therapies. Enrichment of *Akkermansia muciniphila* and *Ruminococcaceae* correlates with improved survival in patients receiving ICIs [42]. Conversely, broad-spectrum antibiotics before immunotherapy reduce treatment efficacy [46]. Preclinical studies suggest that FMT or supplementation with *Akkermansia* can restore ICI responsiveness [29]. Thus, microbiome modulation may emerge as a predictive biomarker and therapeutic tool in HCC management.

### 2.4. Gallbladder Cancer

GBC is aggressive and often associated with gallstone disease. Dysbiosis contributes by altering bile acid metabolism and cholesterol homeostasis, facilitating stone formation [47]. Bacterial taxa such as *Streptococcus* and *Actinomyces* are enriched in GBC tissues [48]. This cancer was selected for its illustration of how microbial shifts in bile and gut ecosystems drive a carcinogenic milieu [49,50].

#### 2.4.1. Microbiome in Oncogenesis

GBC, though less studied, shows strong links to microbial alterations associated with gallstone disease. Dysbiosis increases bile-resistant pathogens (*Enterobacter*, *Klebsiella*) while reducing commensals, resulting in chronic cholecystitis and carcinogenesis [49,50]. Microbes in bile alter cholesterol metabolism, promote lithogenesis, and generate secondary bile acids with DNA-damaging potential [51]. Bacterial biofilms on gallstones facilitate persistent inflammation, providing a carcinogenic niche [52].

#### 2.4.2. Microbiome in Therapy Response

Direct evidence on microbiome and therapy response in GBC is limited. However, microbiome-driven bile acid dysregulation may influence drug metabolism and chemoresistance [53]. Emerging preclinical data suggest that bile microbiome modulation via probiotics could reduce inflammation and improve biliary tract cancer management [54]. Hence, GBC exemplifies bile acid–microbiome–cancer interactions.

### 2.5. Esophageal Cancer

Esophageal adenocarcinoma frequently arises from Barrett’s esophagus, where microbiome shifts occur [55]. Type II microbiota, dominated by Gram-negative anaerobes (*Bacteroides*, *Fusobacteria*), replaces the protective type I community enriched with *Streptococcus* [56]. These changes heighten inflammation and metaplasia, driving progression [57]. Esophageal cancer demonstrates how microbiome alterations in non-colonic regions contribute to carcinogenesis [58].

#### 2.5.1. Microbiome in Oncogenesis

The esophageal microbiome shifts from type I (Gram-positive *Streptococcus*) to type II (Gram-negative anaerobes such as *Prevotella*, *Fusobacterium*) in Barrett’s esophagus [59]. This shift increases LPS-mediated inflammation, nitric oxide generation, and epithelial metaplasia, predisposing to adenocarcinoma [58,59]. In squamous cell carcinoma, enrichment of *Porphyromonas gingivalis* promotes epithelial invasion and immune suppression [60]. Thus, dysbiosis plays a role in both major histological subtypes.

#### 2.5.2. Microbiome in Therapy Response

Esophageal cancer therapy responses are influenced by microbiome composition. Patients with higher microbial diversity respond better to ICI therapy [61]. Dysbiosis may worsen radiation-induced esophagitis, and probiotics have been tested for toxicity reduction [62,63]. These findings highlight esophageal cancer as a site where microbiome alterations influence both pathogenesis and therapeutic tolerance.

### 2.6. Pancreatic Cancer

Pancreatic ductal adenocarcinoma (PDAC) patients display gut dysbiosis with reduced SCFA-producing taxa (*Faecalibacterium*, *Eubacterium*) and increased *Proteobacteria* [64,65]. Metabolites such as trimethylamine *N*-oxide (TMAO) and 3-indoleacetic acid (3-IAA) influence tumor growth and chemotherapy efficacy [66]. As PDAC is notoriously therapy-resistant, microbiome signatures may guide biomarker discovery and novel therapeutic approaches [67,68].

#### 2.6.1. Microbiome in Oncogenesis

PDAC harbors unique microbial signatures, with enrichment of *Pseudomonas* and *Fusobacterium* and depletion of short-chain fatty acid (SCFA) producers (*Faecalibacterium*) [64,65]. Microbial metabolites such as TMAO and 3-IAA promote tumor growth and alter the tumor immune microenvironment [66]. The pancreatic tumor microbiome also suppresses antitumor immunity via myeloid-derived suppressor cell (MDSC) recruitment and TLR activation [69].

#### 2.6.2. Microbiome in Therapy Response

Microbiome contributes to chemoresistance in PDAC. Intra-tumoral *Gammaproteobacteria* can metabolize gemcitabine into inactive derivatives, reducing its efficacy [70]. FMT from long-term survivors into murine models slowed tumor progression and enhanced anti-PD-1 efficacy [71]. Probiotic and dietary interventions to restore SCFA-producing bacteria are under investigation for improving chemotherapy and immunotherapy outcomes [72].

## 3. Extra-Gastrointestinal Cancers

### 3.1. Oral Squamous Cell Carcinoma

Periodontal pathogens including *Porphyromonas gingivalis* and *Fusobacterium nucleatum* promote epithelial proliferation, inflammation, and immune evasion in OSCC [73]. Additionally, *Candida albicans* facilitates tumor invasion and metastasis [74]. This cancer underscores the role of local microbiota beyond the gut, illustrating shared pathogenic mechanisms like chronic inflammation and microbial toxins [75].

#### 3.1.1. Microbiome in Oncogenesis

OSCC pathogenesis is closely linked to oral microbiota. *Porphyromonas gingivalis* inhibits apoptosis, induces epithelial–mesenchymal transition (EMT), and promotes interleukin-6/signal transducer and activator of transcription 3 (IL-6/STAT3) signaling [75]. *Fusobacterium nucleatum* enhances invasion and immune suppression [76]. Additionally, *Candida albicans* contributes to carcinogenesis through nitrosamine production and epithelial disruption [77]. Chronic periodontitis-driven inflammation thus creates a tumor-permissive microenvironment.

#### 3.1.2. Microbiome in Therapy Response

Oral microbiome composition influences radiotherapy- and chemotherapy-induced mucositis severity [78]. Probiotic lozenges with *Lactobacillus* species have shown benefits in reducing mucositis and maintaining oral microbial balance [79]. Studies suggest gut–oral microbial crosstalk may also shape systemic ICI responses, though evidence remains preliminary [80].

### 3.2. Cervical Cancer

Cervical cancer, largely linked to human papillomavirus (HPV), is influenced by vaginal and gut microbiome alterations [81]. Loss of protective *Lactobacillus* and enrichment of anaerobes (e.g., *Gardnerella*, *Mycoplasma*) disrupt mucosal defense and enhance HPV persistence [82]. Evidence suggests that *Mycoplasma* infections further impair p53-mediated tumor suppression [83]. This cancer was included to demonstrate the interplay of microbial dysbiosis with viral oncogenesis [84].

#### 3.2.1. Microbiome in Oncogenesis

Although HPV is the primary driver, microbiome shifts play a cofactor role in cervical cancer. Loss of *Lactobacillus crispatus* and dominance of anaerobes (*Gardnerella*, *Atopobium*, *Mycoplasma*) impair mucosal defense and maintain chronic inflammation [84]. Mycoplasma infections exacerbate genomic instability and hinder p53-mediated DNA repair, facilitating HPV-driven transformation [85]. Dysbiosis thus determines persistence and progression of precancerous lesions.

#### 3.2.2. Microbiome in Therapy Response

Emerging data suggest that vaginal and gut microbiomes influence treatment responses. Lactobacillus dominance correlates with better radiotherapy outcomes, while dysbiosis predicts higher recurrence risk [86]. Probiotic supplementation may improve treatment tolerance, though clinical trials remain limited [87].

### 3.3. Prostate Cancer

Gut dysbiosis is associated with systemic inflammation and metabolic changes that influence prostate carcinogenesis [88]. Mycoplasma species have been implicated in genomic instability and chronic inflammation within the prostate [89]. Emerging studies suggest microbiota may also modulate androgen metabolism and immunotherapy response, justifying its inclusion as a microbiome-associated malignancy outside the GI tract [88].

#### 3.3.1. Microbiome in Oncogenesis

Gut dysbiosis affects systemic inflammation, hormone metabolism, and carcinogenesis in the prostate. Enrichment of *Akkermansia* and depletion of SCFA-producers alter androgen signaling and local immune tone [90]. *Mycoplasma hominis* and other urogenital pathogens have been associated with DNA damage and tumor-promoting inflammation [90]. The prostate thus illustrates potential links between systemic dysbiosis and extra-GI cancer risk, supported primarily by associative and preclinical data.

#### 3.3.2. Microbiome in Therapy Response

Microbiome also influences androgen deprivation therapy (ADT) outcomes [91]. Certain gut taxa can metabolize androgen precursors, potentially contributing to castration resistance [92]. Additionally, gut microbial signatures have been linked to immunotherapy responsiveness in advanced prostate cancer, raising potential for FMT or probiotics as adjuncts [93].

### 3.4. Melanoma

Though not anatomically linked to the gut, melanoma has provided some of the strongest evidence for the role of the microbiome in therapy. Response to PD-1 blockade is significantly associated with gut microbial diversity and enrichment of taxa such as *Akkermansia muciniphila*, *Collinsella aerofaciens*, and *Bifidobacterium longum* [94]. FMT from responders to non-responders restores treatment sensitivity [95]. Melanoma was selected as the most extensively studied model for systemic immune–microbiome interactions in oncotherapy [96].

#### 3.4.1. Microbiome in Oncogenesis

While not directly gut-associated, melanoma progression is influenced by systemic immune regulation shaped by the microbiome. Enrichment of *Bifidobacterium* and *Akkermansia* enhances antigen presentation and CD8^+^ T-cell responses, whereas dysbiosis impairs immune surveillance [97]. The gut microbiome thus indirectly influences melanoma biology through immune modulation.

#### 3.4.2. Microbiome in Therapy Response

Melanoma provides the strongest evidence for microbiome–oncotherapy interaction. Responders to PD-1 blockade consistently show higher gut microbial diversity and abundance of *Akkermansia muciniphila* and *Faecalibacterium* [98]. FMT from ICI responders into non-responders restores treatment sensitivity [12]. Diet and probiotics further modulate therapeutic efficacy, highlighting melanoma as the benchmark cancer for microbiome-immunotherapy research [99].

### 3.5. Brain Tumor

#### 3.5.1. Microbiome in Oncogenesis

Recent evidence suggests that the gut microbiome exerts a significant influence on brain tumor biology through the gut–brain axis, a bidirectional network involving neural, immune, and metabolic signaling [100]. Dysbiosis alters systemic immunity, neuroinflammation, and blood–brain barrier (BBB) integrity—factors increasingly implicated in the progression of gliomas and glioblastoma multiforme (GBM) [101].

Gut microbes produce SCFAs such as butyrate and propionate, which modulate microglial activation and T-cell trafficking within the central nervous system (CNS) [102]. Loss of SCFA-producing taxa (*Faecalibacterium*, *Roseburia*) promotes a pro-inflammatory milieu that facilitates tumor immune evasion [103]. Conversely, pathogenic bacteria and endotoxins can upregulate cytokines such as IL-6 and tumor necrosis factor alpha (TNF-α), enhancing glioma invasiveness [104].

#### 3.5.2. Microbiome in Therapy Response

Preclinical studies show that antibiotic-induced dysbiosis impairs ICI efficacy in glioma models, whereas reconstitution with *Akkermansia muciniphila* or *Bifidobacterium* restores antitumor immunity [105]. Moreover, gut microbial metabolites affect the pharmacokinetics of temozolomide and the local immune tone within the tumor microenvironment [105].

Collectively, these findings support the gut–brain axis as a critical modulator of neuro-oncology, suggesting that microbiome-targeted interventions—diet, probiotics, or fecal microbiota transplantation—may enhance therapeutic efficacy in brain tumors.

### 3.6. Breast Cancer

#### 3.6.1. Microbiome in Oncogenesis

The breast tissue and gut microbiome play crucial roles in breast carcinogenesis through estrogen metabolism, immune modulation, and local inflammation [106]. Dysbiosis in the gut—particularly depletion of *Lactobacillus* and enrichment of *Clostridium* and *Bacteroides*—disrupts the estrobolome, the collection of bacterial genes involved in estrogen metabolism [94]. Increased β-glucuronidase activity leads to reabsorption of active estrogens, promoting hormone-dependent tumor proliferation [107]. In breast tissue, *Methylobacterium radiotolerans* and *Escherichia coli* have been detected more frequently in tumors than in normal tissue, and both can induce DNA double-strand breaks and oxidative stress [107]. Chronic inflammation and immune dysregulation arising from dysbiosis further enhance tumor progression [108].

#### 3.6.2. Microbiome in Therapy Response

Gut microbiota composition affects both chemotherapy tolerance and immunotherapy efficacy [109]. Beneficial taxa such as *Bifidobacterium* and *Akkermansia muciniphila* enhance immune activation and improve responses to ICIs. [94]. Conversely, antibiotic exposure before chemotherapy or immunotherapy reduces progression-free survival in breast cancer patients [110]. Probiotic supplementation and dietary fiber intake have shown promise in improving gut barrier function, mitigating treatment-related mucositis, and potentially enhancing therapeutic outcomes [111].

### 3.7. Lung Cancer

#### 3.7.1. Microbiome in Oncogenesis

The lung and gut microbiomes both play integral roles in LC oncogenesis through inflammatory, metabolic, and immune-mediated pathways [112]. Dysbiosis—characterized by increased *Streptococcus*, *Veillonella*, *Prevotella*, *Enterobacteriaceae*, and *Bacteroides plebeius*—promotes chronic airway inflammation and IL-17/IL-6–driven epithelial proliferation [112]. Reduced abundance of *Faecalibacterium* and *Roseburia*—key butyrate producers—leads to diminished anti-inflammatory signaling, creating a tumor-permissive microenvironment [113].

Recent Mendelian randomization studies have identified causal microbial taxa influencing LC risk. *Coprococcus*, *Holdemanella*, *Peptococcus*, and *Bacteroides clarus* were positively associated with LC susceptibility, while *Collinsella*, *Bifidobacteriaceae*, *Eubacteriaceae*, *Lachnospiraceae UCG-010*, and *Oscillibacter* showed protective associations [114]. Mechanistically, *Collinsella* reduces LC risk by lowering T-cell surface glycoprotein CD5, a regulator of T-cell activation, accounting for approximately 16.7% mediation effect via immune protein modulation. Inflammatory proteins such as IL-20 and IL-8 correlate with increased LC risk, whereas CD5, IL-18, and fibroblast growth factor 21 (FGF21) confer protection, emphasizing the immune-mediated nature of the microbiome–lung axis [115].

#### 3.7.2. Microbiome in Oncotherapy Response

Gut dysbiosis also influences LC oncotherapy outcomes. Enrichment of *Akkermansia muciniphila* and *Ruminococcaceae* is linked to better responses to PD-1/PD-L1 (Programmed death-ligand 1) inhibitors, while antibiotic-induced depletion of beneficial taxa leads to reduced immunotherapy efficacy and shorter progression-free survival [116,117]. Microbiota modulation through diet, probiotics, or FMT represents a promising adjunctive strategy to enhance immunotherapy responsiveness and mitigate treatment-related inflammation [20,118].

The chosen cancers reflect two categories, GI malignancies where gut microbes directly interact with the tumor microenvironment, and extra-GI malignancies where microbiome-driven systemic inflammation or immune modulation plays a pivotal role. Together, they provide a balanced framework to explore both oncogenesis mechanisms and oncotherapy implications across cancer types, fulfilling the rationale for a broad-spectrum narrative review.

## 4. Discussion

The interplay between the gut microbiome and cancer has emerged as a central theme in oncology, spanning carcinogenesis, tumor progression, and therapeutic modulation. This review of thirteen cancers highlights both common pathways of microbiome involvement and unique axes of interaction, while also underscoring the translational opportunities for diagnostics and therapeutics.

### 4.1. Common Mechanistic Themes Across Cancers

Across GI and extra-GI cancers, several shared mechanisms define the microbiome–oncogenesis link.

Chronic inflammation is perhaps the most consistent of all events. Microbial products such as LPS and flagellin engage TLRs, activating NF-κB and STAT3 signaling pathways, which promote cytokine release, epithelial proliferation, and immune evasion [119]. This is evident in CRC, esophageal, and oral cancers, where *Fusobacterium nucleatum* and *Porphyromonas gingivalis* drive pro-inflammatory signaling that fosters tumorigenesis (Figure 4) [120]. Genotoxic metabolites represent another recurrent mechanism. Colibactin-producing *Escherichia coli* induces double-strand DNA breaks in CRC, while nitrosating bacteria in gastric cancer generate N-nitroso compounds, potent mutagens that damage epithelial DNA (Figure 5) [121]. In cervical cancer, microbial dysbiosis impairs antiviral immunity, allowing HPV persistence and increased genomic instability [122]. Barrier dysfunction is also a cross-cutting theme. Dysbiosis reduces butyrate-producing commensals such as *Faecalibacterium prausnitzii*, weakening epithelial tight junctions and allowing microbial translocation [123]. This is especially relevant in HCC, where increased gut permeability delivers bacterial endotoxins directly to the liver via the portal vein, perpetuating inflammation and carcinogenesis [124]. Finally, immune modulation serves as a unifying principle in cancer–microbiome crosstalk. Beneficial taxa such as *Akkermansia muciniphila* and *Bifidobacterium longum* enhance antigen presentation and CD8^+^ T-cell activation, whereas pathogenic species such as *Fusobacterium* suppress NK cell activity and recruit myeloid-derived suppressor cells (MDSCs) that blunt antitumor immunity [111].

### 4.2. Unique Microbiome–Cancer Axes

While these broad mechanisms are shared, certain cancers illustrate unique axes of the microbiome influence. In HCC, the gut–liver axis is central. Dysbiosis alters bile acid metabolism, disrupting FXR/TGR5 signaling and increasing carcinogenic secondary bile acids [125]. Endotoxin leakage further drives hepatic inflammation and fibrosis, distinguishing HCC from other cancers [126]. In OSCC, the oral microbiome is the primary driver. Periodontal pathogens such as *P. gingivalis* not only induce local inflammation but also manipulate host cell signaling by activating the phosphoinositide 3-kinase (PI3K)/Akt pathway and inhibiting apoptosis [127]. *Candida albicans* further contributes by generating carcinogenic nitrosamines, demonstrating the unique role of fungal dysbiosis [128]. In melanoma, the microbiome’s role is systemic rather than local. Here, gut microbial diversity and the presence of immunostimulatory taxa predict response to ICIs [129]. This represents the clearest example of the microbiome influencing extra-GI cancers through systemic immune priming [38].

**Figure 4 cancers-18-00099-f004:**
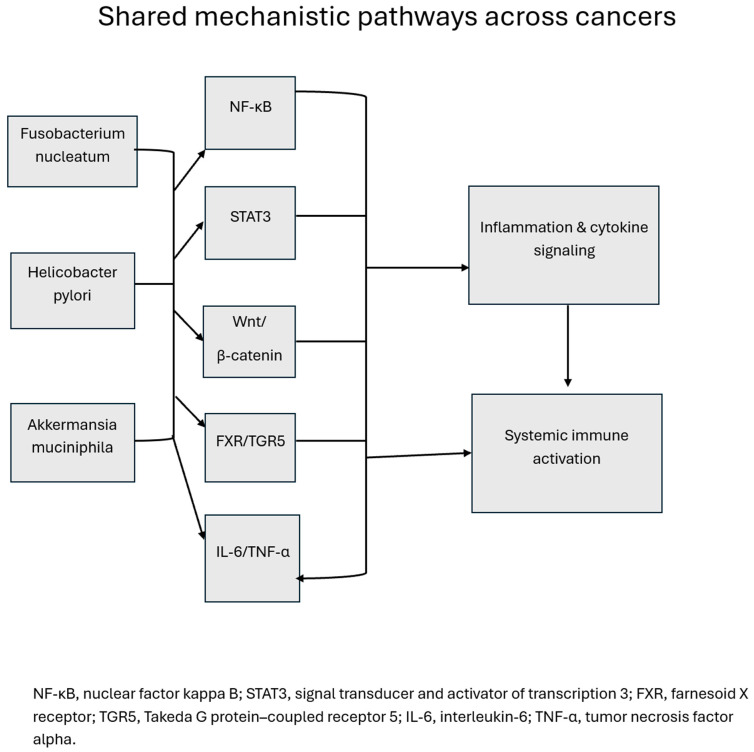
Shared mechanistic pathways across cancers. Representative microbial taxa (*Fusobacterium nucleatum*, *Helicobacter pylori*, *Akkermansia muciniphila*) activate conserved oncogenic and inflammatory signaling cascades including nuclear factor kappa b (NF-κB), signal transducer and activator of transcription 3 (STAT3), Wnt/β-catenin, farnesoid X receptor; Takeda G-protein-coupled-receptor 5 (FXR/TGR5), and Interleukin 6/tumor necrosis factor alpha (IL-6/TNF-α) [130]. These overlapping networks drive inflammation, cytokine release, and systemic immune activation across multiple malignancies.

### 4.3. Clinical Translation and Therapeutic Implications

The translational potential of microbiome research is substantial, spanning biomarkers, preventive strategies, and therapeutic adjuncts. From a clinical standpoint, current microbiome-related applications in oncology should be interpreted with caution. At present, the strongest evidence supports avoidance of unnecessary broad-spectrum antibiotic exposure around immune checkpoint inhibitor therapy, given its consistent association with impaired treatment response [131]. In contrast, interventions such as fecal microbiota transplantation, engineered probiotics, or targeted dietary modulation remain investigational and should be limited to clinical trial settings. Routine microbiome profiling is not yet recommended for clinical decision-making outside research protocols [132].

**Figure 5 cancers-18-00099-f005:**
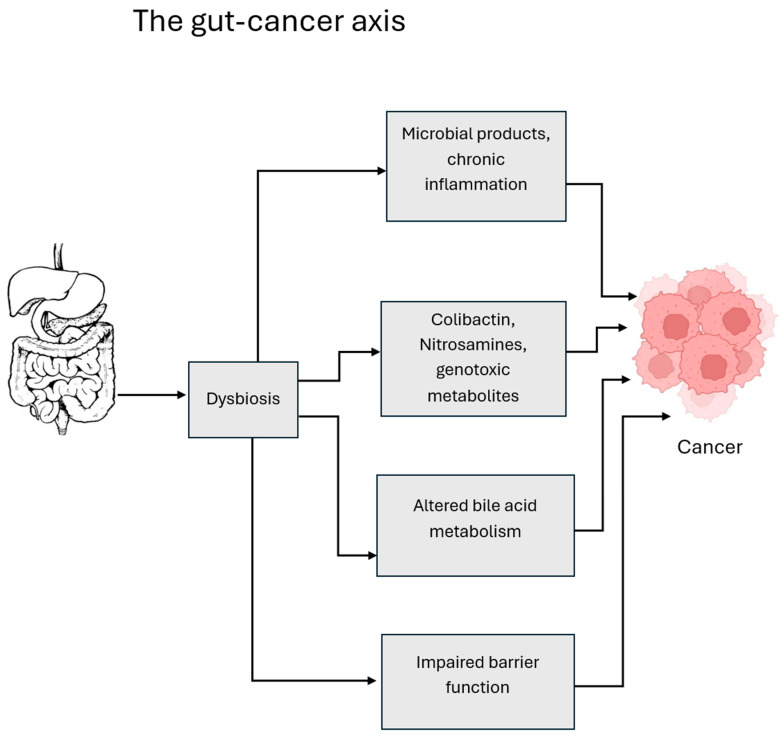
This schematic summarizes key mechanisms linking gut dysbiosis to oncogenesis. Microbial imbalance promotes chronic inflammation, production of genotoxic metabolites such as colibactin and nitrosamines, bile-acid dysregulation with impaired FXR/TGR5 signaling, and disruption of the intestinal barrier [121]. These convergent pathways foster genomic instability and tumor initiation [133].

### 4.4. Microbiome as a Biomarker

Microbial signatures are increasingly proposed as predictive biomarkers for cancer risk and therapy response. For instance, *Fusobacterium nucleatum* abundance in CRC tissue correlates with poor prognosis and chemoresistance, while the presence of *Akkermansia* predicts better outcomes with ICI therapy in HCC and melanoma [134,135]. Incorporating microbiome profiling into oncology practice could aid in risk stratification and personalized therapy planning.

Diet profoundly shapes the microbiome. High fiber intake increases SCFA production, which enhances epithelial barrier function and supports antitumor immunity [72]. Observational studies link dietary fiber with improved immunotherapy outcomes in melanoma [136]. Probiotics such as *Lactobacillus rhamnosus* and *Bifidobacterium* have been tested in gastric and oral cancers to reduce treatment toxicity and improve mucosal health, though evidence remains preliminary [137].

Fecal microbiota transplantation (FMT) is a promising but still experimental therapeutic approach, with early studies demonstrating the restoration of immunotherapy responsiveness in previously resistant melanoma patients [138]. Pilot studies suggest similar potential in CRC and HCC, though standardization of donor selection, delivery methods, and long-term safety remains a challenge. The future lies in precision approaches using targeted prebiotics, engineered probiotics, or small-molecule modulators to selectively enhance beneficial taxa while suppressing pathogenic species. Integration of microbiome sequencing with host genomic and immunologic profiling could allow personalized oncotherapy regimens tailored to individual microbial ecosystems.

## 5. Conclusions

The gut microbiome has emerged as a critical determinant in both oncogenesis and response to cancer therapies. Accumulating evidence demonstrates that microbial communities influence tumor initiation, progression, and metastasis through multiple mechanisms, including modulation of inflammation, immune surveillance, genotoxic metabolite production, and epithelial barrier integrity. In addition, the gut microbiota significantly impacts the efficacy and toxicity of oncotherapies, including chemotherapy, immunotherapy, and targeted treatments, highlighting its role as a potential predictive biomarker and therapeutic target. For instance, specific microbial signatures are associated with (CRC) development, while modulation of the gut microbiome enhances checkpoint inhibitor response in melanoma and LC (Table 1). Beyond direct tumor effects, the microbiome also influences systemic metabolism and immune tone, further affecting therapy outcomes.

Looking forward, precision manipulation of the microbiome holds considerable promise in oncology. Strategies such as dietary interventions, probiotics, prebiotics, FMT, and microbial metabolite modulation may complement existing treatments, improving efficacy while reducing adverse effects. However, translating these insights into clinical practice requires rigorous longitudinal studies, mechanistic validation, and standardized protocols. Integrating microbiome profiling into personalized cancer care could pave the way for next-generation, microbiome-informed oncotherapy, offering the potential for improved patient outcomes and novel preventive strategies. Overall, the microbiome represents a transformative frontier in understanding and managing cancer.

## Figures and Tables

**Figure 1 cancers-18-00099-f001:**
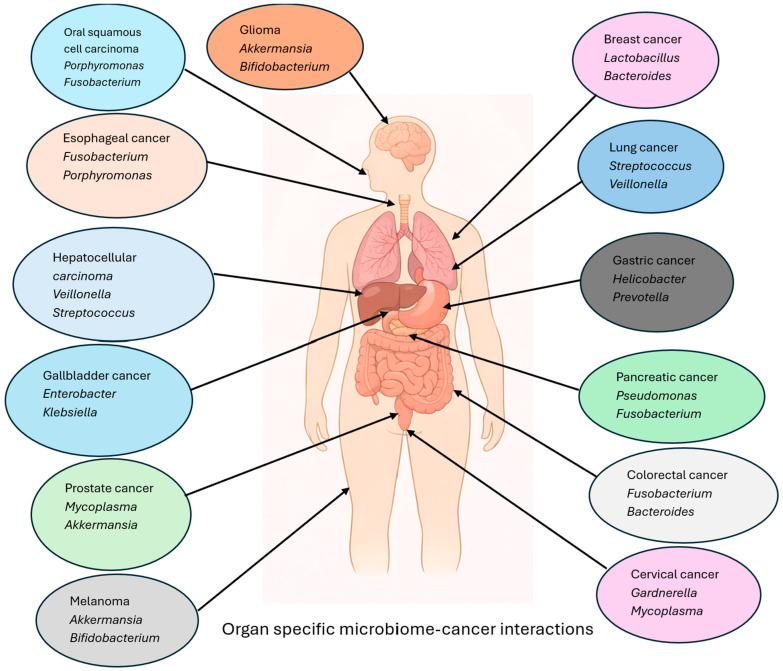
Organ-specific associations between the microbiome and cancer types. Schematic illustration summarizing representative microbial taxa reported to be associated with thirteen cancer types across gastrointestinal and extra-gastrointestinal organs.

**Figure 2 cancers-18-00099-f002:**
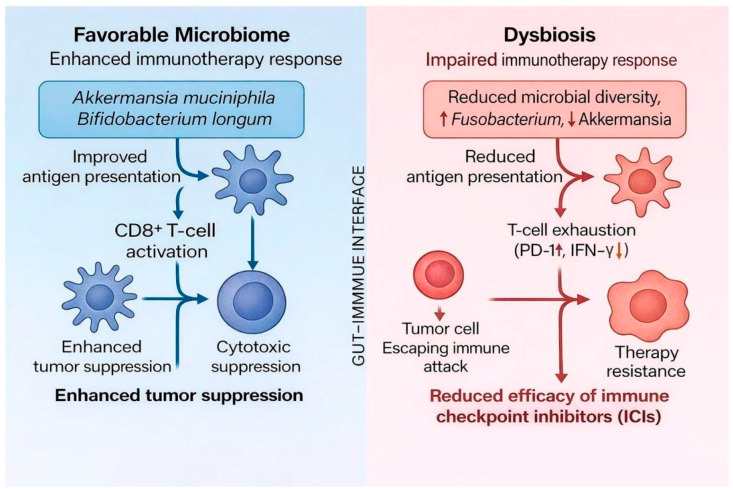
Microbiome influence on immunotherapy response. A schematic summarizing how gut microbial composition modulates the efficacy of ICIs. A favorable microbiome enriched with *Akkermansia muciniphila* and *Bifidobacterium longum* enhances antigen presentation and CD8^+^ T-cell activation, promoting cytotoxic tumor suppression [15]. Conversely, dysbiosis characterized by reduced diversity and overgrowth of *Fusobacterium* or loss of *Akkermansia* leads to impaired antigen presentation, T-cell exhaustion (↑ PD-1, ↓ IFN-γ), and diminished ICI efficacy [16].

**Figure 3 cancers-18-00099-f003:**
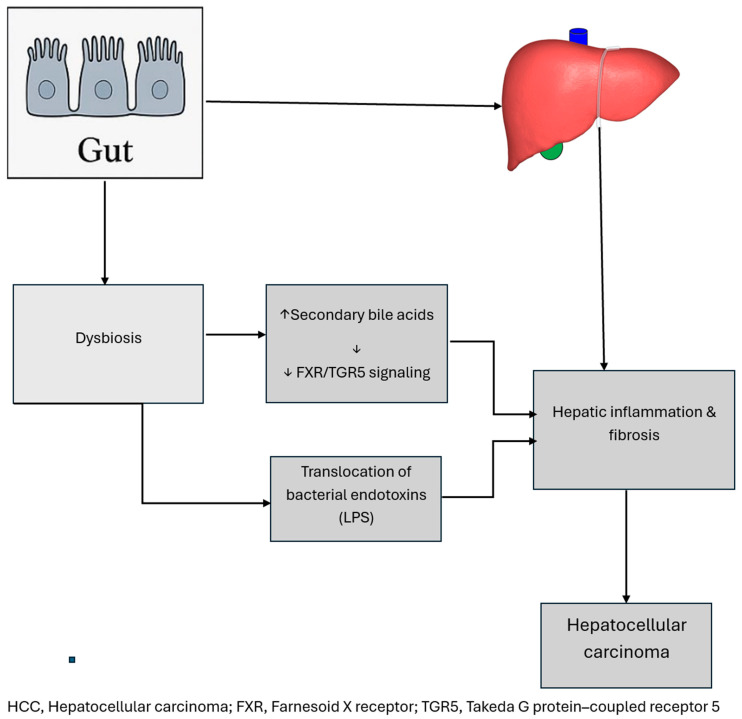
Gut–liver axis in hepatocellular carcinoma. Gut dysbiosis increases secondary bile acids that suppress hepatic Farnesoid X Receptor (FXR) and (Takeda G protein-coupled receptor (TGR5) signaling, resulting in inflammation, fibrosis, and hepatocellular carcinoma (HCC) [43]. The figure highlights the bidirectional gut–liver axis mediated by portal circulation and bile-acid feedback loops. HCC, Hepatocellular; FXR, Farnesoid X receptor; TGR5, Takeda G protein-coupled receptor 5.

**Table 1 cancers-18-00099-t001:** Comparative summary of gut microbiome in oncogenesis and oncotherapies across thirteen cancer types.

Cancer Type	Microbes	Oncogenesis Mechanism	Therapy Modulation	Clinical Implications
Colorectal cancer (CRC)	*Fusobacterium nucleatum*, *Bacteroides fragilis*, *Escherichia coli*	Wnt/β-catenin activation, DNA damage, inflammation	Resistance to 5-FU/oxaliplatin; ICI response linked to *Akkermansia*	Microbial profiling may guide chemo-immunotherapy strategies
Gastric cancer	*Helicobacter pylori*, *Prevotella*, *Neisseria*	CagA/VacA-induced DNA damage; nitrosating bacteria generate carcinogens	Alters chemotherapy efficacy; probiotics reduce toxicity	Eradication plus microbiome support may lower cancer risk
Hepatocellular carcinoma (HCC)	*Veillonella*, *Streptococcus*, *Akkermansia*	Bile acid dysregulation, LPS-driven inflammation	ICI outcomes linked to *Akkermansia* enrichment	Microbiome as biomarker for immunotherapy response
Gallbladder cancer	*Enterobacter*, *Klebsiella*, *Streptococcus*	Bile acid imbalance, gallstone biofilms, chronic inflammation	Limited evidence; bile dysbiosis may influence drug metabolism	Potential role of probiotics in biliary cancer prevention
Esophageal cancer	*Fusobacterium*, *Porphyromonas*, *Prevotella*	Dysbiosis in Barrett’s esophagus, inflammation, nitric oxide generation	Microbial diversity predicts ICI response; probiotics mitigate radiation toxicity	Microbiome may serve as risk marker and therapeutic adjunct
Pancreatic cancer	*Pseudomonas*, *Fusobacterium*, *Gammaproteobacteria*	SCFA loss; TMAO/3-IAA promote growth and immunosuppression	*Gammaproteobacteria* metabolize gemcitabine; FMT improves ICI efficacy	Microbial modulation may overcome chemoresistance
Oral SCC	*Porphyromonas gingivalis*, *Fusobacterium nucleatum*, *Candida*	Inflammation, EMT induction, nitrosamine production	Oral probiotics reduce mucositis; may support systemic therapy	Oral–gut microbial axis relevant for prevention and therapy
Cervical cancer	*Gardnerella*, *Mycoplasma*, *Atopobium*	Dysbiosis impairs HPV clearance, genomic instability	Vaginal microbiome influences radiotherapy outcomes	Microbiome restoration could reduce recurrence risk
Prostate cancer	*Mycoplasma*, *Akkermansia*, *SCFA-producing taxa*	Inflammation, DNA damage, altered androgen metabolism	Gut microbes modulate ADT and ICI outcomes	Microbiome-targeted therapies may delay resistance
Melanoma	*Akkermansia*, *Bifidobacterium*, *Faecalibacterium*	Immune modulation, enhanced T-cell activation	ICI efficacy linked to microbial diversity; FMT restores response	Benchmark cancer for microbiome–immunotherapy translation
Glioma/glioblastoma (brain tumors)	*Akkermansia muciniphila*, *Bifidobacterium*, *Faecalibacterium prausnitzii*, *Roseburia*, *Escherichia coli* (*LPS-producing*), *Clostridium* spp.	Dysbiosis reduces SCFA-producing taxa (e.g., Faecalibacterium, Roseburia), weakening anti-inflammatory signaling and disrupting the gut–brain axis. Bacterial metabolites and endotoxins cross a compromised gut barrier, inducing systemic inflammation, IL-6/TNF-α release, and microglial activation that promotes tumor proliferation and immune escape	Antibiotic-induced dysbiosis impairs ICI efficacy; reintroduction of *Akkermansia* or *Bifidobacterium* restores T-cell activation and response. SCFAs modulate microglial phenotype and BBB integrity, influencing temozolomide metabolism and local immune tone	Gut–brain axis modulation via probiotics, prebiotics, or FMT may enhance ICI response and chemotherapy effectiveness; microbial biomarkers could help predict treatment sensitivity and neuroinflammation risk
Breast cancer	*Lactobacillus*, *Bacteroides*, *Clostridium*, *Methylobacterium radiotolerans*, *Escherichia coli*, *Bifidobacterium*	Gut dysbiosis alters estrobolome activity → increased β-glucuronidase → higher circulating estrogens; local bacteria (*E. coli*, Methylobacterium) induce DNA breaks and oxidative stress; immune modulation via pro-inflammatory signaling	*Akkermansia muciniphila* and *Bifidobacterium* enhance ICI efficacy; antibiotics impair chemo-/immunotherapy response; probiotics improve mucosal repair	Microbial profiling may identify hormone-responsive risk; probiotic and dietary fiber interventions could enhance treatment efficacy and reduce toxicity
Lung cancer	*Streptococcus*, *Veillonella*, *Prevotella*, *Bacteroides*, *Akkermansia*, *Ruminococcaceae*	Chronic airway and systemic inflammation; IL-17/IL-6–driven epithelial proliferation; reduced SCFA-producing taxa leading to impaired anti-inflammatory signaling	Gut microbial diversity and enrichment of *Akkermansia* and *Ruminococcaceae* associated with improved ICI response; antibiotic-induced dysbiosis reduces immunotherapy efficacy	Microbiome profiling may predict immunotherapy response and guide antibiotic stewardship during ICI treatment

Abbreviations: SCC, squamous cell carcinoma; CagA, cytotoxin-associated gene A; VacA, vacuolating cytotoxin A; LPS, lipopolysaccharide; SCFA, short-chain fatty acid; TMAO, trimethylamine-N-oxide; 3-IAA, 3-indoleacetic acid; EMT, epithelial–mesenchymal transition; HPV, human papillomavirus; IL-6, interleukin-6; TNF-α, tumor necrosis factor alpha; 5-FU, 5-fluorouracil; ICI, immune checkpoint inhibitor; FMT, fecal microbiota transplantation; ADT, androgen deprivation therapy; BBB, blood–brain barrier.

## Data Availability

No new data were created or analyzed in this study.

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
