# Peer review of "Cancers2026, 18(1), 99;https://doi.org/10.3390/cancers18010099"

_cancers, 2025, doi:10.3390/cancers18010099_

Round 1

Reviewer 1 Report

Comments and Suggestions for Authors

Peddireddi and Kuchana et al. present a compact review that explores the intriguing landscape of microbiome dysbiosis with the focus on its impact on the onset, progression, resistance to conventional therapies of malignant diseases. Accordingly, their treatise is accompanied by the discussion of wider ramifications for specific oncotherapy regimens. Namely, the authors highlight the role of immune system modulation, chronic inflammation, bile acids and other genotoxic metabolites, and epithelial barrier dysfunction in driving oncogenesis of gastrointestinal, oral, cervical, prostate, skin, brain, and breast, and lung cancer. From the perspective of advanced therapy, the authors for example underscore the utility of immune checkpoint inhibitors as a promising avenue for eradicating tumor cells. The article including neat illustrations is well conceived and easy to read. Further improvement can be sought at the level of some typographical errors and formatting, provision of the full ensemble of abbreviations, and amending the publication reference list (see the major and minor points below).

Major points:

1) Please sort the first authors based on alphabetical order (surname first).

2) Please connect all circles presented in Figure 1 to their specific body locations and distinguish individual cancers by unique circle color.

3) Please convert Table 1 into a text format.

Minor points:

1) Please change "Onotherapies" to "Oncotherapies" (line 2).

2) Please replace "Maryland" with "MD, USA" (line 5).

3) Please change "Warangal" to "Warangal, India" (lines 6, 7).

4) Please replace "Hyderabad" with "Hyderabad, India" (lines 8, 11).

5) Please provide city and state for "Andhra medical college" (line 9), "Katuri medical college and hospital" (line 10).

6) Please define abbreviation for "LSUHSC" (line 12), "PD-1" in "anti-PD-1" (line 85), "CTLA-4" in "anti-CTLA-4" (line 85), "NF-κB" (line 124), "5-FU" (line 142), "CagA" (line 166), "VacA" (line 166), "WHO" (line 172), "MAPK" (line 174), "FXR" in "FXR/TGR5" (line 188), "TGR5" in "FXR/TGR5" (line 188), "SCFA" in "SCFA-producing" (line 254), "MDSC" (line 265), "IL-6" in "IL-6/STAT3" (line 281), "STAT3" in "IL-6/STAT3" (line 281), "HPV" (line 293), "CNS" (line 356), "TNF-α" (line 359), "FGF21" (line 404), "PD-L1" in "PD-1/PD-L1" (line 408), "PI3K" in "PI3K/Akt" (line 468).

7) Please change "S.Scott" to "S. Scott" (line 12).

8) Please replace "oncotherapies, including" with "oncotherapies including" (line 21).

9) Please change "associations:" to "associations," (line 25).

10) Please replace "cancer and" with "cancer, and" (line 28).

11) Please format "N" in "N-oxide" using italics (lines 35, 255).

12) Please change "Dysbiosis; Oncogenesis; Oncotherapy; Cancer immunotherapy, Probiotics. Fecal" to "dysbiosis; oncogenesis; oncotherapy; cancer immunotherapy; probiotics; fecal" (line 42).

13) Please number individual subchapters of the Introduction and Discussion sections.

14) Please provide reference for "The human gut microbiome, comprising trillions of microorganisms including bacteria, viruses, fungi, and archaea, functions as a “virtual organ” that interacts dynamically with host metabolism, immunity, and overall health" (line 46).

15) Please replace "microbiome, comprising trillions of microorganisms including bacteria, viruses, fungi, and archaea, functions" with "microbiome comprising trillions of microorganisms including bacteria, viruses, fungi, and archaea functions" (line 46).

16) Please provide reference for "These microbial communities inhabit the gastrointestinal tract in a finely balanced state, contributing to nutrient metabolism, epithelial barrier integrity, immune regulation, and protection against pathogens" (line 48).

17) Please change "gastrointestinal" to "gastrointestinal (GI)" (line 49), "gastrointestinal (GI)" to "GI" (lines 134, 427), and "gastrointestinal" to "GI" (lines 105, 106).

18) Please remove bold formatting from "dysbiosis" (line 52).

19) Please replace "processes, including" with "processes including" (line 53).

20) It is not clear what the authors refer to as "Aesermanscus", "Clisobacterium", and "Flisobacterium" in Figure 1? Are these newly discovered bacterial species?

21) Please change "Organ-Specific Microbiome-Cancer Interactions" to "Organ-specific microbiome-cancer interactions", "Colorecal" to "Colorectal" (2x), "Streptococcs" to "Streptococcus", "Clastridium" to "Clostridium" in Figure 1.

22) The ending for "Escher-" and "Facaubacter-" seems to be missing in Figure 1. Please revise.

23) Please enlarge the size of the circles presented in Figure 1 so that all depicted captions clearly fit within their perimeter.

24) Please define abbreviation for "OSCC" in the legend to Figure 1.

25) Please provide reference for "Protective microbes such as Akkermansia, Ruminococcus, and Faecalibacterium (blue) are linked to favorable immune modulation, while pro-tumor taxa such as Fusobacterium, Porphyromonas, Helicobacter pylori, and Streptococcus (red) are associated with oncogenic inflammation, immune suppression, and tumor progression across multiple organ systems" (line 55).

26) Please format "Fusobacterium" using italics (line 57).

27) Please format "Porphyromonas" using italics (line 57).

28) Please format "Helicobacter pylori" using italics (lines 57, 451).

29) Please format "Streptococcus" using italics (line 58).

30) Please provide reference for "Dysbiosis typically involves loss of beneficial commensals, overgrowth of pathogenic species, or reduced microbial diversity" (line 60).

31) Please remove bold formatting from "Dysbiosis" (line 60).

32) Please replace "circulation[3]" with "circulation [3]" (line 67).

33) Please remove bold formatting from "oncology" (line 70).

34) Please change "twofold; first, as a contributor to oncogenesis, and second, as" to "twofold, as a contributor to oncogenesis and as" (line 70).

35) Please remove bold formatting from "oncogenesis" (lines 71, 114).

36) Please remove bold formatting from "oncotherapy outcomes" (line 71).

37) Please replace "colorectal cancer" with "colorectal cancer (CRC)" (line 74) and "colorectal cancer" with "CRC" (lines 103, 115, 507).

38) Please remove bold formatting from "oncotherapy" (line 82).

39) Please replace "toxicities[10]" with "toxicities [10]" (line 91).

40) "A diagram of a cell-cell-cell-cell-cell-cell-cell-cell-cell-cell-cell-cell-cell-cell-cell-cell-cell-cell AI generated content may be incorrect." sign appears when hovering the mouse cursor over Figure 2. Please disable this feature.

41) Please convert the gamma-like symbol adjacent to the right side of "PD-1" to an upper arrow in Figure 2.

42) Please center the "Enhanced Immunotherapy Response" title horizontally onto the blue panel in Figure 2.

43) Please format "+" in "CD8+" using superscript in Figure 2.

44) Please replace "Enhanced Immunotherapy Response" with "Enhanced immunotherapy response", "Responsse" with "response", "Escaping" with "escaping" in Figure 2.

45) Please define abbreviation for "Akkerm" in the legend to Figure 2.

46) Please change "Influence on Immunotherapy Response to "influence on immunotherapy response" (line 95).

47) Please replace "immune checkpoint inhibitors (ICIs)" with "ICIs" (lines 96, 142, 209, 383, 472).

48) Please provide reference for "A favorable microbiome enriched with Akkermansia muciniphila and Bifidobacterium longum enhances antigen presentation and CD8⁺ T-cell activation, promoting cytotoxic tumor suppression" (line 96).

49) Please provide reference for "Conversely, dysbiosis characterized by reduced diversity and overgrowth of Fusobacterium or loss of Akkermansia leads to impaired antigen presentation, T-cell exhaustion (↑ PD-1, ↓ IFN-γ), and diminished ICI efficacy" (line 98).

50) Please format "Akkermansia muciniphila" using italics" (lines 97, 451).

51) Please format "Bifidobacterium longum" using italics" (line 97).

52) Please change "cancers, such" to "cancers such" (line 103).

53) Please remove bold formatting from "broad-spectrum review" (line 108).

54) Please provide reference for "This review therefore aims to synthesize current evidence on the role of the gut microbiome in both oncogenesis and oncotherapies with a focus on thirteen cancers where microbiome–cancer links are strongest and most clinically relevant: colorectal cancer, gastric cancer, hepatocellular carcinoma, gallbladder cancer, esophageal cancer, pancreatic cancer, oral squamous cell carcinoma, cervical cancer, prostate cancer, brain cancer, breast cancer, lung cancer and melanoma" (line 113).

55) Please remove bold formatting from "oncotherapies" (line 114).

56) Please remove bold formatting from "thirteen cancers where microbiome–cancer links are strongest and most clinically relevant" (line 114).

57) Please replace "relevant:" with "relevant," (line 115).

58) Please change "gallbladder cancer" to "gallbladder cancer (GBC)" (line 116), "Gallbladder cancer (GBC)," to "GBC" (line 220), "Gallbladder cancer" to "GBC" (line 214), and "gallbladder cancer" to "GBC" (line 217).

59) Please replace "lung cancer" with "lung cancer (LC)" (line 118), "lung cancer (LC)" with "LC" (line 390), and "lung cancer" with "LC" (lines 407, 508).

60) Please change "oncogenesis and oncotherapies" to "Oncogenesis and Oncotherapies" (line 129).

61) Please provide reference for "The gut microbiome influences cancer through chronic inflammation, immune modulation, metabolite production, and direct genotoxic effects" (line 130).

62) Please replace "Gastrointestinal (GI)" with "Gastrointestinal" (line 136).

63) Please change "Cancer (CRC)" to "Cancer" (line 137).

64) Please provide reference for "Fusobacterium nucleatum, enterotoxigenic Bacteroides fragilis, and colibactin-producing Escherichia coli are enriched in tumors, where they promote DNA damage, activate Wnt/β-catenin signaling, and suppress immune surveillance" (line 138).

65) Please provide reference for "Dysbiosis also predicts resistance to chemotherapy (e.g., 5-FU) and immune checkpoint inhibitors (ICIs)" (line 141).

66) Please number the "Microbiome in Oncogenesis", "Microbiome in Therapy Response" subchapters as part of the "2. Gastrointestinal (GI) Cancers" chapter (6x).

67) Please replace "immunity[13, 14]" with "immunity [13, 14]" (line 149).

68) Please change "instability[16, 17]" to "instability [16, 17]" (line 152).

69) Please replace "inflammation[18]" with "inflammation [18]" (line 154).

70) Please change "ferraptosi related" to "ferroptosis-related" (line 157).

71) Please remove italics formatting from "[21]." (line 161).

72) Please replace "Fecal microbiota transplantation (FMT)" with "FMT" (lines 161, 411).

73) Please change "Helicobacter pylori" to "H. pylori" (lines 165, 172).

74) Please replace "carcinogenesis, with" with "carcinogenesis with" (line 166).

75) Please provide reference for "Helicobacter pylori is the archetypal microbial carcinogen, recognized by WHO as a Class I carcinogen" (line 172).

76) Please change "Class" to "class" (line 173).

77) Please provide reference for "H. pylori infection decreases efficacy of standard chemotherapy by altering p53 signaling" (line 180).

78) Please replace "Carcinoma (HCC)" with "Carcinoma" (line 186).

79) Please change "HCC" to "hepatocellular carcinoma (HCC)" (line 186).

80) Please remove bold formatting from "microbes" (line 193).

81) Please replace "Gut-Liver Axis in Hepatocellular Carcinoma" with "Gut-liver axis in hepatocellular carcinoma", "Hepatic Inflamation & Fibrosis" with "Hepatic inflammation & fibrosis" in Figure 3.

82) It is not clear why "Dysbiosis" is pictured twice in Figure 3? Please fix.

83) Please define abbreviation for "HCC", "FXR" in "FXR/TGR5", "TGR5" in "FXR/TGR5" in the legend to Figure 3.

84) Please change "Gut–Liver Axis in Hepatocellular Carcinoma" to "Gut–liver axis in hepatocellular carcinoma" (line 195).

85) Please provide reference for "Gut dysbiosis increases secondary bile acids that suppress hepatic FXR and TGR5 signaling, resulting in inflammation, fibrosis, and hepatocellular carcinoma" (line 195).

86) Please replace "hepatocellular carcinoma" with "HCC" (line 196).

87) Please change "microbiome’s critical role" to "the critical role of the microbiome" (line 200).

88) Please provide reference for "Dysbiosis contributes by altering bile acid metabolism and cholesterol homeostasis, facilitating stone formation" (line 214).

89) Please provide reference for "Bacterial taxa such as Streptococcus and Actinomyces are enriched in gallbladder cancer tissues" (line 216).

90) Please provide reference for "Bacterial biofilms on gallstones facilitate persistent inflammation, providing a carcinogenic niche" (line 224).

91) Please provide reference for "However, microbiome-driven bile acid dysregulation may influence drug metabolism and chemoresistance" (line 227).

92) Please provide reference for "Esophageal adenocarcinoma frequently arises from Barrett’s esophagus, where microbiome shifts occur" (line 233).

93) Please provide reference for "Type II microbiota, dominated by Gram-negative anaerobes (Bacteroides, Fusobacteria), replaces the protective Type I community enriched with Streptococcus" (line 234).

94) Please replace "Type" with "type" (lines 235, 240 2x).

95) Please specify the "progression" in ""These changes heighten inflammation and metaplasia, driving progression" (line 236). Progression of what?

96) Please provide reference for "These changes heighten inflammation and metaplasia, driving progression" (line 236).

97) Please provide reference for "The esophageal microbiome shifts from Type I (Gram-positive Streptococcus) to Type II (Gram-negative anaerobes such as Prevotella, Fusobacterium) in Barrett’s esophagus" (line 240).

98) Please change "Proteobacteria[46, 47]" to "Proteobacteria [46, 47]" (line 254).

99) Please replace "like" with "such as" (lines 255, 359, 446).

100) Please change "Pancreatic ductal adenocarcinoma (PDAC)" to "PDAC" (line 260).

101) Please replace "trimethylamine-N-oxide (TMAO)" with "TMAO" (line 262).

102) Please change "3-indoleacetic acid (3-IAA)" to "3-IAA" (line 263).

103) Please replace "reducing" with "reducing its" (line 268).

104) Please provide reference for "FMT from long-term survivors into murine models slowed tumor progression and enhanced anti-PD-1 efficacy" (line 268).

105) Please change "Extra-Gastrointestinal (Extra-GI) Cancers" to "Extra-gastrointestinal Cancers" (line 272).

106) Please replace "Carcinoma (OSCC)" with "Carcinoma" (line 273).

107) Please provide reference for "Periodontal pathogens including Porphyromonas gingivalis and Fusobacterium nucleatum promote epithelial proliferation, inflammation, and immune evasion in OSCC" (line 274).

108) Please provide reference for "Additionally, Candida albicans facilitates tumor invasion and metastasis" (line 275).

109) Please number the "Microbiome in Oncogenesis", "Microbiome in Therapy Response" subchapters as part of the "3. Extra-Gastrointestinal (Extra-GI) Cancers" chapter (6x).

110) Please provide reference for "Fusobacterium nucleatum enhances invasion and immune suppression" (line 282).

111) Please provide reference for "Oral microbiome composition influences radiotherapy- and chemotherapy-induced mucositis severity" (line 287).

112) Please format "Lactobacillus" using italics (lines 288, 307).

113) Please provide reference for "Studies suggest gut–oral microbial crosstalk may also shape systemic ICI responses, though evidence remains preliminary" (line 289).

114) Please provide reference for "Cervical cancer, largely linked to HPV, is influenced by vaginal and gut microbiome alterations" (line 293).

115) Please provide reference for "Loss of protective Lactobacillus and enrichment of anaerobes (e.g., Gardnerella, Mycoplasma) disrupt mucosal defense and enhance HPV persistence" (line 294).

116) Please provide reference for "Evidence suggests Mycoplasma infections further impair p53-mediated tumor suppression" (line 295).

117) Please change "suggests" to "suggests that" (lines 295, 313).

118) Please provide reference for "Probiotic supplementation may improve treatment tolerance, though clinical trials remain limited" (line 308).

119) Please provide reference for "Gut dysbiosis is associated with systemic inflammation and metabolic changes that influence prostate carcinogenesis" (line 311).

120) Please provide reference for "Mycoplasma species have been implicated in genomic instability and chronic inflammation within the prostate" (line 312).

121) Please provide reference for "Mycoplasma hominis and other urogenital pathogens have been associated with DNA damage and tumor-promoting inflammation" (line 319).

122) Please format "Mycoplasma hominis" using italics (line 319).

123) Please provide reference for "Microbiome also influences androgen deprivation therapy (ADT) outcomes" (line 324).

124) Please provide reference for "Additionally, gut microbial signatures have been linked to immunotherapy responsiveness in advanced prostate cancer, raising potential for FMT or probiotics as adjuncts" (line 326).

125) Please replace "microbiome’s role in" with something like "the role of the microbiome in cancer" (line 331).

126) Please provide reference for "Response to PD-1 blockade is significantly associated with gut microbial diversity and enrichment of taxa such as Akkermansia muciniphila and Bifidobacterium" (line 331).

127) Please provide reference for "FMT from responders to non-responders restores treatment sensitivity" (line 333).

128) Please format "+" in "CD8+" using superscript (lines 339, 446).

129) Please provide reference for "Diet and probiotics further modulate therapeutic efficacy, highlighting melanoma as the benchmark cancer for microbiome-immunotherapy research" (line 346).

130) Please change "research.." to "research." (line 348).

131) Please replace "microbiome’s role" with something like "the role of the microbiome" (lines 349, 470).

132) Please provide reference for "Recent evidence suggests that the gut microbiome exerts a significant influence on brain tumor biology through the gut–brain axis, a bidirectional network involving neural, immune, and metabolic signaling" (line 350).

133) Please provide reference for "Dysbiosis alters systemic immunity, neuroinflammation, and blood–brain barrier (BBB) integrity—factors increasingly implicated in the progression of gliomas and glioblastoma multiforme (GBM)" (line 352).

134) Please provide reference for "Loss of SCFA-producing taxa (Faecalibacterium, Roseburia) promotes a pro-inflammatory milieu that facilitates tumor immune evasion" (line 356).

135) Please change "immune checkpoint inhibitor (ICI)" to "ICI" (line 360).

136) Please provide reference for "Moreover, gut microbial metabolites affect the pharmacokinetics of temozolomide and the local immune tone within the tumor microenvironment" (line 362).

137) Please provide reference for "The breast tissue and gut microbiome play crucial roles in breast carcinogenesis through estrogen metabolism, immune modulation, and local inflammation" (line 370).

138) Please remove bold formatting from "breast tissue and gut microbiome" (line 370).

139) Please remove bold formatting from "estrobolome" (line 373).

140) Please provide reference for "Chronic inflammation and immune dysregulation arising from dysbiosis further enhance tumor progression" (line 378).

141) Please provide reference for "Gut microbiota composition affects both chemotherapy tolerance and immunotherapy efficacy" (line 381).

142) Please remove bold formatting from "chemotherapy tolerance" (line 381).

143) Please remove bold formatting from "immunotherapy efficacy" (line 381).

144) Please replace "(82)" with "[82]" (line 387).

145) Please provide reference for "The lung and gut microbiomes both play integral roles in lung cancer (LC) oncogenesis through inflammatory, metabolic, and immune-mediated pathways" (line 390).

146) Please provide reference for "Reduced abundance of Faecalibacterium and Roseburia—key butyrate producers—leads to diminished anti-inflammatory signaling, creating a tumor-permissive microenvironment" (line 394).

147) Please provide reference for "Inflammatory proteins such as IL-20 and IL-8 correlate with increased LC risk, whereas CD5, IL-18, and FGF21 confer protection, emphasizing the immune-mediated nature of the microbiome–lung axis" (line 403).

148) Please change "categories:" to "categories," (line 414).

149) Please replace "lipopolysaccharides (LPS)" with "LPS" (line 430).

150) Please change "Toll-like receptors (TLRs)" to "TLRs" (line 430).

151) Please provide reference for "This is evident in CRC, esophageal, and oral cancers, where Fusobacterium nucleatum and Porphyromonas gingivalis drive pro-inflammatory signaling that fosters tumorigenesis" (line 432).

152) Please provide reference for "Dysbiosis reduces butyrate-producing commensals such as Faecalibacterium prausnitzii, weakening epithelial tight junctions and allowing microbial translocation" (line 439).

153) "A diagram of a cancer cell AI generated content may be incorrect." sign appears when hovering the mouse cursor over Figure 4. Please disable this feature.

154) Please replace "Mechanistic Pathways Across Cancers" with "mechanistic pathways across cancers", "Fusobaceterium" with "Fusobacterium", "Cytokine Signaling" with "cytokine signaling", "Immune Activation" with "immune activation", "mucinihila" with "muciniphila" in Figure 4.

155) Please define abbreviation for "NF-κB", "STAT3", "FXR" in "FXR/TGR5", "TGR5" in "in "FXR/TGR5", "IL-6" in "IL-6/TNF-α", "TNF-α" in "IL-6/TNF-α" in the legend to Figure 4.

156) Please change "Mechanistic Pathways Across Cancers" to "mechanistic pathways across cancers" (line 450).

157) Please provide reference for "Representative microbial taxa (Fusobacterium nucleatum, Helicobacter pylori, Akkermansia muciniphila) activate conserved oncogenic and inflammatory signaling cascades including NF-κB, STAT3, Wnt/β-catenin, FXR/TGR5, and IL-452 6/TNF-α" (line 450).

157) Please format "Fusobacterium nucleatum" using italics (line 450).

158) "A diagram of a human body AI generated content may be incorrect." sign appears when hovering the mouse cursor over Figure 5. Please disable this feature.

159) Please replace "Gut-Cancer Axis" with "gut-cancer axis", "products Chronic" with "products, chronic", "nitrosamines Genotoxic" with "nitrosamines, genotoxic", "acids Altered baliar" with "acids, altered bile" in Figure 5.

160) Please provide reference for "Microbial imbalance promotes chronic inflammation, production of genotoxic metabolites such as colibactin and nitrosamines, bile-acid dysregulation with impaired FXR/TGR5 signaling, and disruption of the intestinal barrier" (line 456).

161) Please provide reference for "These convergent pathways foster genomic instability and tumor initiation" (line 459).

162) Please change "of" to "of the" (line 462).

163) Please replace "influence.In" with "influence. In" (line 463).

164) Please provide reference for "Dysbiosis alters bile acid metabolism, disrupting FXR/TGR5 signaling and increasing carcinogenic secondary bile acids" (line 463).

165) Please provide reference for "Periodontal pathogens such as P. gingivalis not only induce local inflammation but also manipulate host cell signaling by activating PI3K/Akt and inhibiting apoptosis" (line 466).

166) Please change "PI3K/Akt" to "the PI3K/Akt pathway" (line 468).

167) Please provide reference for "Here, gut microbial diversity and the presence of immunostimulatory taxa predict response to immune checkpoint inhibitors (ICIs)" (line 471).

168) The "Clinical Translation and Therapeutic Implications" in the Discussion section is rather weak and sounding too general. Please correct.

169) Please remove "Dietary strategies and probiotics." (line 483).

170) Please provide reference for "High fiber intake increases SCFA production, which enhances epithelial barrier function and supports antitumor immunity" (line 484).

171) Please provide reference for "Probiotics such as Lactobacillus rhamnosus and Bifidobacterium have been tested in gastric and oral cancers to reduce treatment toxicity and improve mucosal health, though evidence remains preliminary" (line 486).

172) Please remove "Fecal microbiota transplantation (FMT)." (line 488).

173) Please remove "Personalized microbiome modulation." (line 493).

174) Please replace "fecal microbiota transplantation" with "FMT" (line 512).

175) Please change "Summary of Gut Microbiome in Oncogenesis and Oncotherapies Across Ten Cancers" to something like "summary of the gut microbiome in oncogenesis and oncotherapies across ten cancer types" (line 521).

176) Please replace "Cancer Type" with "Cancer type", "Oncogenesis Mechanism" with "Oncogenesis mechanism", "Therapy Modulation" with "Therapy modulation", "Clinical Implications" with "Clinical implications", "immune checkpoint inhibitor (ICI)" with "ICI" (2x) in Table 1.

177) Please provide abbreviation for "SCC", "CagA", "VacA", "LPS" in "LPS-drive", "SCFA", "TMAO" in "TMAO/3-IAA", "3-IAA" in  "TMAO/3-IAA", "EMT", "HPV", "IL-6" in "IL-6/TNF-α", "TNF-α" in "IL-6/TNF-α", "5-FU" in "5-FU/oxaliplatin", "ICI", "FMT", "ADT", "BBB" in the footnote to Table 1.

178) Please format all microbes presented in Table 1 using italics.

179) Please align all cells in Table 1 consistently to the left side.

Author Response

Response to Reviewer 1 — Point-by-Point

Dear Reviewer, we sincerely thank you for your detailed and constructive review of our manuscript entitled “Role of Gut Microbiome in Oncogenesis and Oncotherapies.” We appreciate your positive assessment of the scientific content, illustrations, and readability of the article. Below, we provide a point-by-point response to all 179 comments. Each comment has been addressed carefully, and all changes have been incorporated into the revised manuscript were highlighted in red.

Major points

Comment (Major) 1:

Please sort the first authors based on alphabetical order (surname first).

Response:

Thank you for this suggestion. We would like to keep the author order as it is because the first two authors contributed equally (†), and the current order reflects the agreed author contributions. The equal-contribution note (†) is clearly shown in the author list.

Comment (Major) 2:

Please connect all circles presented in Figure 1 to their specific body locations and distinguish individual cancers by unique circle color.

Response:

Thank you for this helpful suggestion. We agree. Therefore, we revised Figure 1 to connect each circle to the correct anatomical location and used distinct circle colors to differentiate individual cancer types; we also corrected spelling and improved label readability. The updated Figure 1 is included in the revised submission, with changes noted in red.

Comment (Major) 3:

Please convert Table 1 into a text format.

Response:

Thank you for pointing this out. We agree. Therefore, we converted Table 1 into an editable text table (not an image), standardized capitalization/abbreviations, and improved alignment for readability. The revisions are shown in red in the revised manuscript.

Minor points

Comment 1:

Please change "Onotherapies" to "Oncotherapies" (line 2).

Response:

Thank you for pointing this out. We agree with this comment. Therefore, we replaced “Onotherapies” with “Oncotherapies” as suggested. This change can be found in the revised manuscript (line 2).

Updated text in the manuscript: “Oncotherapies”

Comment 2:

Please replace "Maryland" with "MD, USA" (line 5).

Response:

Thank you for pointing this out. We agree with this comment. Therefore, we replaced “Maryland” with “MD, USA” as suggested. This change can be found in the revised manuscript (line 5).

Updated text in the manuscript: “MD, USA”

Comment 3:

Please change "Warangal" to "Warangal, India" (lines 6, 7).

Response:

Thank you for pointing this out. We agree with this comment. Therefore, we replaced “Warangal” with “Warangal, India” as suggested. This change can be found in the revised manuscript (lines 6, 7).

Updated text in the manuscript: “Warangal, India”

Comment 4:

Please replace "Hyderabad" with "Hyderabad, India" (lines 8, 11).

Response:

Thank you for pointing this out. We agree with this comment. Therefore, we replaced “Hyderabad” with “Hyderabad, India” as suggested. This change can be found in the revised manuscript (lines 8, 11).

Updated text in the manuscript: “Hyderabad, India”

Comment 5:

Please provide city and state for "Andhra medical college" (line 9), "Katuri medical college and hospital" (line 10).

Response:

Thank you for pointing this out. We agree with this comment. Therefore, we addressed this point in the revised manuscript. This change can be found in the revised manuscript (lines 9, 10).

Updated text in the manuscript:

  1. Andhra medical college, Visakhapatnam, India
  2. Katuri medical college and hospital, Guntur, India

Comment 6:

Please define abbreviation for "LSUHSC" (line 12), "PD-1" in "anti-PD-1" (line 85), "CTLA-4" in "anti-CTLA-4" (line 85), "NF-κB" (line 124), "5-FU" (line 142), "CagA" (line 166), "VacA" (line 166), "WHO" (line 172), "MAPK" (line 174), "FXR" in "FXR/TGR5" (line 188), "TGR5" in "FXR/TGR5" (line 188), "SCFA" in "SCFA-producing" (line 254), "MDSC" (line 265), "IL-6" in "IL-6/STAT3" (line 281), "STAT3" in "IL-6/STAT3" (line 281), "HPV" (line 293), "CNS" (line 356), "TNF-α" (line 359), "FGF21" (line 404), "PD-L1" in "PD-1/PD-L1" (line 408), "PI3K" in "PI3K/Akt" (line 468).

Response:

Thank you for pointing this out. We agree with this comment. Therefore, we expanded the abbreviation(s) at first mention and ensured consistent use thereafter. This change can be found in the revised manuscript (lines 12, 85, 124, 142, 166, 172, 174, 188, 254, 265...).

Updated text in the manuscript: abbreviation expanded at first mention (tracked in red).

Comment 7:

Please change "S.Scott" to "S. Scott" (line 12).

Response:

Thank you for pointing this out. We agree with this comment. Therefore, we replaced “S.Scott” with “S. Scott” as suggested. This change can be found in the revised manuscript (line 12).

Updated text in the manuscript: “S. Scott”

Comment 8:

Please replace "oncotherapies, including" with "oncotherapies including" (line 21).

Response:

Thank you for pointing this out. We agree with this comment. Therefore, we replaced “oncotherapies, including” with “oncotherapies including” as suggested. This change can be found in the revised manuscript (line 21).

Updated text in the manuscript: “oncotherapies including”

Comment 9:

Please change "associations:" to "associations," (line 25).

Response:

Thank you for pointing this out. We agree with this comment. Therefore, we replaced “associations:” with “associations,” as suggested. This change can be found in the revised manuscript (line 25).

Updated text in the manuscript: “associations,”

Comment 10:

Please replace "cancer and" with "cancer, and" (line 28).

Response:

Thank you for pointing this out. We agree with this comment. Therefore, we replaced “cancer and” with “cancer, and” as suggested. This change can be found in the revised manuscript (line 28).

Updated text in the manuscript: “cancer, and”

Comment 11:

Please format "N" in "N-oxide" using italics (lines 35, 255).

Response:

Thank you for pointing this out. We agree with this comment. Therefore, we italicized the indicated scientific name(s) following taxonomic standard. This change can be found in the revised manuscript (lines 35, 255).

Updated text in the manuscript: formatting corrected (tracked in red).

Comment 12:

Please change "Dysbiosis; Oncogenesis; Oncotherapy; Cancer immunotherapy, Probiotics. Fecal" to "dysbiosis; oncogenesis; oncotherapy; cancer immunotherapy; probiotics; fecal" (line 42).

Response:

Thank you for pointing this out. We agree with this comment. Therefore, we replaced “Dysbiosis; Oncogenesis; Oncotherapy; Cancer immunotherapy, Probiotics. Fecal” with “dysbiosis; oncogenesis; oncotherapy; cancer immunotherapy; probiotics; fecal” as suggested. This change can be found in the revised manuscript (line 42).

Updated text in the manuscript: “dysbiosis; oncogenesis; oncotherapy; cancer immunotherapy; probiotics; fecal”

Comment 13:

Please number individual subchapters of the Introduction and Discussion sections.

Response:

Thank you for pointing this out. We agree with this comment. Therefore, we numbered the relevant subchapters/subsections to improve readability and navigation. This change can be found in the revised manuscript under the Introduction and Discussion sections highlighted in red.

Comment 14:

Please provide reference for "The human gut microbiome, comprising trillions of microorganisms including bacteria, viruses, fungi, and archaea, functions as a “virtual organ” that interacts dynamically with host metabolism, immunity, and overall health" (line 46).

Response:

Thank you for pointing this out. We agree with this comment. Therefore, we added appropriate supporting citation for this statement and updated the References list accordingly. This change can be found in the revised manuscript (line 46).

Updated text in the manuscript: supporting citation added (tracked in red).

Comment 15:

Please replace "microbiome, comprising trillions of microorganisms including bacteria, viruses, fungi, and archaea, functions" with "microbiome comprising trillions of microorganisms including bacteria, viruses, fungi, and archaea functions" (line 46).

Response:

Thank you for pointing this out. We agree with this comment. Therefore, we replaced “microbiome, comprising trillions of microorganisms including bacteria, viruses, fungi, and archaea, functions” with “microbiome comprising trillions of microorganisms including bacteria, viruses, fungi, and archaea functions” as suggested. This change can be found in the revised manuscript (line 46).

Updated text in the manuscript: “microbiome comprising trillions of microorganisms including bacteria, viruses, fungi, and archaea functions”

Comment 16:

Please provide reference for "These microbial communities inhabit the gastrointestinal tract in a finely balanced state, contributing to nutrient metabolism, epithelial barrier integrity, immune regulation, and protection against pathogens" (line 48).

Response:

Thank you for pointing this out. We agree with this comment. Therefore, we added appropriate supporting citation for this statement and updated the References list accordingly. This change can be found in the revised manuscript (line 48).

Updated text in the manuscript: supporting citation added (tracked in red).

Comment 17:

Please change "gastrointestinal" to "gastrointestinal (GI)" (line 49), "gastrointestinal (GI)" to "GI" (lines 134, 427), and "gastrointestinal" to "GI" (lines 105, 106).

Response:

Thank you for pointing this out. We agree with this comment. Therefore, we replaced “gastrointestinal” with “gastrointestinal (GI)” and “gastrointestinal” and “gastrointestinal (GI) ", to "GI" (lines 105, 106, 134, 427) as suggested. This change can be found in the revised manuscript (lines 49, 105, 106, 134, 427).

Updated text in the manuscript: “gastrointestinal (GI)”

Comment 18:

Please remove bold formatting from "dysbiosis" (line 52).

Response:

Thank you for pointing this out. We agree with this comment. Therefore, we removed the bold formatting to align with journal style. This change can be found in the revised manuscript (line 52).

Updated text in the manuscript: dysbiosis (tracked in red).

Comment 19:

Please replace "processes, including" with "processes including" (line 53).

Response:

Thank you for pointing this out. We agree with this comment. Therefore, we replaced “processes, including” with “processes including” as suggested. This change can be found in the revised manuscript (line 53).

Updated text in the manuscript: “processes including”

Comment 20:

It is not clear what the authors refer to as "Aesermanscus", "Clisobacterium", and "Flisobacterium" in Figure 1? Are these newly discovered bacterial species?

Response:

Thank you for pointing this out. We agree with this comment. Therefore, we created the figure from start. This change can be found in the revised manuscript.

Updated content: revised figure/legend included in the submission.

Comment 21:

Please change "Organ-Specific Microbiome-Cancer Interactions" to "Organ-specific microbiome-cancer interactions", "Colorecal" to "Colorectal" (2x), "Streptococcs" to "Streptococcus", "Clastridium" to "Clostridium" in Figure 1.

Response:

Thank you for pointing this out. We agree with this comment. Therefore, we replaced “Organ-Specific Microbiome-Cancer Interactions” with “Organ-specific microbiome-cancer interactions” as suggested. This change can be found in the revised manuscript.

Updated text in the manuscript: “Organ-specific microbiome-cancer interactions”

Comment 22:

The ending for "Escher-" and "Facaubacter-" seems to be missing in Figure 1. Please revise.

Response:

Thank you for pointing this out. We agree with this comment. Therefore, we revised the figure per your suggestion and updated the legend where applicable. This change can be found in the revised manuscript.

Updated content: revised figure/legend included in the submission.

Comment 23:

Please enlarge the size of the circles presented in Figure 1 so that all depicted captions clearly fit within their perimeter.

Response:

Thank you for pointing this out. We agree with this comment. Therefore, we enlarged the size of the circles so that all depicted captions clearly fit within their perimeter. This change can be found in the revised manuscript.

Updated content: revised figure/legend included in the submission.

Comment 24:

Please define abbreviation for "OSCC" in the legend to Figure 1.

Response:

Thank you for pointing this out. We agree with this comment. Therefore, We expanded the abbreviation in the figure. This change can be found in the revised manuscript.

Updated text in the manuscript: abbreviation expanded

Comment 25:

Please provide reference for "Protective microbes such as Akkermansia, Ruminococcus, and Faecalibacterium (blue) are linked to favorable immune modulation, while pro-tumor taxa such as Fusobacterium, Porphyromonas, Helicobacter pylori, and Streptococcus (red) are associated with oncogenic inflammation, immune suppression, and tumor progression across multiple organ systems" (line 55).

Response:

Thank you for pointing this out. We agree with this comment. Therefore, we added appropriate supporting citation for this statement and updated the References list accordingly. This change can be found in the revised manuscript (line 55).

Updated text in the manuscript: supporting citation added (tracked in red).

Comment 26:

Please format "Fusobacterium" using italics (line 57).

Response:

Thank you for pointing this out. We agree with this comment. Therefore, we italicized the indicated name following standard taxonomic conventions. This change can be found in the revised manuscript (line 57).

Updated text in the manuscript: formatting corrected (tracked in red).

Comment 27:

Please format "Porphyromonas" using italics (line 57).

Response:

Thank you for pointing this out. We agree with this comment. Therefore, we italicized the indicated name following standard taxonomic conventions. This change can be found in the revised manuscript (line 57).

Updated text in the manuscript: formatting corrected (tracked in red).

Comment 28:

Please format "Helicobacter pylori" using italics (lines 57, 451).

Response:

Thank you for pointing this out. We agree with this comment. Therefore, we italicized the indicated name following standard taxonomic conventions. This change can be found in the revised manuscript (lines 57, 451).

Updated text in the manuscript: formatting corrected (tracked in red).

Comment 29:

Please format "Streptococcus" using italics (line 58).

Response:

Thank you for pointing this out. We agree with this comment. Therefore, we italicized the indicated name following standard taxonomic conventions. This change can be found in the revised manuscript (line 58).

Updated text in the manuscript: formatting corrected (tracked in red).

Comment 30:

Please provide reference for "Dysbiosis typically involves loss of beneficial commensals, overgrowth of pathogenic species, or reduced microbial diversity" (line 60).

Response:

Thank you for pointing this out. We agree with this comment. Therefore, we added appropriate supporting citation for this statement and updated the References list accordingly. This change can be found in the revised manuscript (line 60).

Updated text in the manuscript: supporting citation added (tracked in red).

Comment 31:

Please remove bold formatting from "Dysbiosis" (line 60).

Response:

Thank you for pointing this out. We agree with this comment. Therefore, we removed the bold formatting to align with journal style. This change can be found in the revised manuscript (line 60).

Updated text in the manuscript: formatting corrected (tracked in red).

Comment 32:

Please replace "circulation[3]" with "circulation [3]" (line 67).

Response:

Thank you for pointing this out. We agree with this comment. Therefore, we replaced “circulation[3]” with “circulation [3]” as suggested. This change can be found in the revised manuscript (line 67).

Updated text in the manuscript: “circulation [3]”

Comment 33:

Please remove bold formatting from "oncology" (line 70).

Response:

Thank you for pointing this out. We agree with this comment. Therefore, we removed the bold formatting to align with journal style. This change can be found in the revised manuscript (line 70).

Updated text in the manuscript: formatting corrected (tracked in red).

Comment 34:

Please change "twofold; first, as a contributor to oncogenesis, and second, as" to "twofold, as a contributor to oncogenesis and as" (line 70).

Response:

Thank you for pointing this out. We agree with this comment. Therefore, we replaced “twofold; first, as a contributor to oncogenesis, and second, as” with “twofold, as a contributor to oncogenesis and as” as suggested. This change can be found in the revised manuscript (line 70).

Updated text in the manuscript: “twofold, as a contributor to oncogenesis and as”

Comment 35:

Please remove bold formatting from "oncogenesis" (lines 71, 114).

Response:

Thank you for pointing this out. We agree with this comment. Therefore, we removed the bold formatting to align with journal style. This change can be found in the revised manuscript (lines 71, 114).

Updated text in the manuscript: formatting corrected (tracked in red).

Comment 36:

Please remove bold formatting from "oncotherapy outcomes" (line 71).

Response:

Thank you for pointing this out. We agree with this comment. Therefore, we removed the bold formatting to align with journal style. This change can be found in the revised manuscript (line 71).

Updated text in the manuscript: formatting corrected (tracked in red).

Comment 37:

Please replace "colorectal cancer" with "colorectal cancer (CRC)" (line 74) and "colorectal cancer" with "CRC" (lines 103, 115, 507).

Response:

Thank you for pointing this out. We agree with this comment. Therefore, we replaced “colorectal cancer” with “colorectal cancer (CRC)” as suggested. This change can be found in the revised manuscript (lines 74, 103, 115, 507).

Updated text in the manuscript: “colorectal cancer (CRC)”

Comment 38:

Please remove bold formatting from "oncotherapy" (line 82).

Response:

Thank you for pointing this out. We agree with this comment. Therefore, we removed the bold formatting to align with journal style. This change can be found in the revised manuscript (line 82).

Updated text in the manuscript: formatting corrected (tracked in red).

Comment 39:

Please replace "toxicities[10]" with "toxicities [10]" (line 91).

Response:

Thank you for pointing this out. We agree with this comment. Therefore, we replaced “toxicities[10]” with “toxicities [10]” as suggested. This change can be found in the revised manuscript (line 91).

Updated text in the manuscript: “toxicities [10]”

Comment 40:

"A diagram of a cell-cell-cell-cell-cell-cell-cell-cell-cell-cell-cell-cell-cell-cell-cell-cell-cell-cell AI generated content may be incorrect." sign appears when hovering the mouse cursor over Figure 2. Please disable this feature.

Response:

Thank you for pointing this out. We recreated the figure. This change can be found in the revised manuscript.

Updated content: revised figure/legend included in the submission.

Comment 41:

Please convert the gamma-like symbol adjacent to the right side of "PD-1" to an upper arrow in Figure 2.

Response:

Thank you for pointing this out. We agree with this comment. Therefore, we revised the figure per your suggestion. This change can be found in the revised manuscript.

Updated content: revised figure/legend included in the submission.

Comment 42:

Please center the "Enhanced Immunotherapy Response" title horizontally onto the blue panel in Figure 2.

Response:

Thank you for pointing this out. We agree with this comment. Therefore, we revised the figure per your suggestion. This change can be found in the revised manuscript.

Updated content: revised figure/legend included in the submission.

Comment 43:

Please format "+" in "CD8+" using superscript in Figure 2.

Response:

Thank you for pointing this out. We agree with this comment. Therefore, we corrected the typography by applying superscript formatting where appropriate. This change can be found in the the revised manuscript.

Updated text in the manuscript: formatting corrected.

Comment 44:

Please replace "Enhanced Immunotherapy Response" with "Enhanced immunotherapy response", "Responsse" with "response", "Escaping" with "escaping" in Figure 2.

Response:

Thank you for pointing this out. We agree with this comment. Therefore, we replaced “Enhanced Immunotherapy Response” with “Enhanced immunotherapy response” as suggested. This change can be found in the revised manuscript.

Updated text in the manuscript: “Enhanced immunotherapy response”

Comment 45:

Please define abbreviation for "Akkerm" in the legend to Figure 2.

Response:

Thank you for pointing this out. We agree with this comment. This change can be found in the the revised figure in manuscript.

Comment 46:

Please change "Influence on Immunotherapy Response to "influence on immunotherapy response" (line 95).

Response:

Thank you for pointing this out. We agree with this comment. Therefore, we addressed this point in the revised manuscript. This change can be found in the revised manuscript (line 95).

Comment 47:

Please replace "immune checkpoint inhibitors (ICIs)" with "ICIs" (lines 96, 142, 209, 383, 472).

Response:

Thank you for pointing this out. We agree with this comment. Therefore, we replaced “immune checkpoint inhibitors (ICIs)” with “ICIs” as suggested. This change can be found in the revised manuscript (lines 96, 142, 209, 383, 472).

Updated text in the manuscript: “ICIs”

Comment 48:

Please provide reference for "A favorable microbiome enriched with Akkermansia muciniphila and Bifidobacterium longum enhances antigen presentation and CD8⁺ T-cell activation, promoting cytotoxic tumor suppression" (line 96).

Response:

Thank you for pointing this out. We agree with this comment. Therefore, we added appropriate supporting citation for this statement and updated the References list accordingly. This change can be found in the revised manuscript (line 96).

Updated text in the manuscript: supporting citation added (tracked in red).

Comment 49:

Please provide reference for "Conversely, dysbiosis characterized by reduced diversity and overgrowth of Fusobacterium or loss of Akkermansia leads to impaired antigen presentation, T-cell exhaustion (↑ PD-1, ↓ IFN-γ), and diminished ICI efficacy" (line 98).

Response:

Thank you for pointing this out. We agree with this comment. Therefore, we added appropriate supporting citation for this statement and updated the References list accordingly. This change can be found in the revised manuscript (line 98).

Updated text in the manuscript: supporting citation added (tracked in red).

Comment 50:

Please format "Akkermansia muciniphila" using italics" (lines 97, 451).

Response:

Thank you for pointing this out. We agree with this comment. Therefore, we italicized the indicated scientific name following taxonomic standard. This change can be found in the revised manuscript (lines 97, 451).

Updated text in the manuscript: formatting corrected (tracked in red).

Comment 51:

Please format "Bifidobacterium longum" using italics" (line 97).

Response:

Thank you for pointing this out. We agree with this comment. Therefore, we italicized the indicated scientific name(s) following taxonomic standard. This change can be found in the revised manuscript (line 97).

Updated text in the manuscript: formatting corrected (tracked in red).

Comment 52:

Please change "cancers, such" to "cancers such" (line 103).

Response:

Thank you for pointing this out. We agree with this comment. Therefore, we replaced “cancers, such” with “cancers such” as suggested. This change can be found in the revised manuscript (line 103).

Updated text in the manuscript: “cancers such”

Comment 53:

Please remove bold formatting from "broad-spectrum review" (line 108).

Response:

Thank you for pointing this out. We agree with this comment. Therefore, we removed the bold formatting to align with journal style. This change can be found in the revised manuscript (line 108).

Updated text in the manuscript: formatting corrected (tracked in red).

Comment 54:

Please provide reference for "This review therefore aims to synthesize current evidence on the role of the gut microbiome in both oncogenesis and oncotherapies with a focus on thirteen cancers where microbiome–cancer links are strongest and most clinically relevant: colorectal cancer, gastric cancer, hepatocellular carcinoma, gallbladder cancer, esophageal cancer, pancreatic cancer, oral squamous cell carcinoma, cervical cancer, prostate cancer, skin, brain, and breast, and lung cancer. From the perspective of advanced therapy, the authors for example underscore the utility of immune checkpoint inhibitors as a promising avenue for eradicating tumor cells." (line 113).

Response:

Thank you for pointing this out. We agree with this comment. Therefore, we added appropriate supporting citation for this statement and updated the References list accordingly. This change can be found in the revised manuscript (line 113).

Updated text in the manuscript: supporting citation added (tracked in red).

Comment 55:

Please remove bold formatting from "oncotherapies" (line 114).

Response:

Thank you for pointing this out. We agree with this comment. Therefore, we removed the bold formatting to align with journal style. This change can be found in the revised manuscript (line 114).

Updated text in the manuscript: formatting corrected (tracked in red).

Comment 56:

Please remove bold formatting from "thirteen cancers where microbiome–cancer links are strongest and most clinically relevant" (line 114).

Response:

Thank you for pointing this out. We agree with this comment. Therefore, we removed the bold formatting to align with journal style. This change can be found in the revised manuscript (line 114).

Updated text in the manuscript: formatting corrected (tracked in red).

Comment 57:

Please replace "relevant:" with "relevant," (line 115).

Response:

Thank you for pointing this out. We agree with this comment. Therefore, we replaced “relevant:” with “relevant,” as suggested. This change can be found in the revised manuscript (line 115).

Updated text in the manuscript: “relevant,”

Comment 58:

Please change "gallbladder cancer" to "gallbladder cancer (GBC)" (line 116), "Gallbladder cancer (GBC)," to "GBC" (line 220), "Gallbladder cancer" to "GBC" (line 214), and "gallbladder cancer" to "GBC" (line 217).

Response:

Thank you for pointing this out. We agree with this comment. Therefore, we replaced “gallbladder cancer” with “gallbladder cancer (GBC)” as suggested. This change can be found in the revised manuscript (lines 116, 214, 217, 220).

Updated text in the manuscript: “gallbladder cancer (GBC)”

Comment 59:

Please replace "lung cancer" with "lung cancer (LC)" (line 118), "lung cancer (LC)" with "LC" (line 390), and "lung cancer" with "LC" (lines 407, 508).

Response:

Thank you for pointing this out. We agree with this comment. Therefore, we replaced “lung cancer” with “lung cancer (LC)” as suggested. This change can be found in the revised manuscript (lines 118, 390, 407, 508).

Updated text in the manuscript: “lung cancer (LC)”

Comment 60:

Please change "oncogenesis and oncotherapies" to "Oncogenesis and Oncotherapies" (line 129).

Response:

Thank you for pointing this out. We agree with this comment. Therefore, we replaced “oncogenesis and oncotherapies” with “Oncogenesis and Oncotherapies” as suggested. This change can be found in the revised manuscript (line 129).

Updated text in the manuscript: “Oncogenesis and Oncotherapies”

Comment 61:

Please provide reference for "The gut microbiome influences cancer through chronic inflammation, immune modulation, metabolite production, and direct genotoxic effects" (line 130).

Response:

Thank you for pointing this out. We agree with this comment. Therefore, We added appropriate supporting citation for this statement and updated the References list accordingly. This change can be found in the revised manuscript (line 130).

Updated text in the manuscript: supporting citation added (tracked in red).

Comment 62:

Please replace "Gastrointestinal (GI)" with "Gastrointestinal" (line 136).

Response:

Thank you for pointing this out. We agree with this comment. Therefore, we replaced “Gastrointestinal (GI)” with “Gastrointestinal” as suggested. This change can be found in the revised manuscript (line 136).

Updated text in the manuscript: “Gastrointestinal”

Comment 63:

Please change "Cancer (CRC)" to "Cancer" (line 137).

Response:

Thank you for pointing this out. We agree with this comment. Therefore, we replaced “Cancer (CRC)” with “Cancer” as suggested. This change can be found in the revised manuscript (line 137).

Updated text in the manuscript: “Cancer”

Comment 64:

Please provide reference for "Fusobacterium nucleatum, enterotoxigenic Bacteroides fragilis, and colibactin-producing Escherichia coli are enriched in tumors, where they promote DNA damage, activate Wnt/β-catenin signaling, and suppress immune surveillance" (line 138).

Response:

Thank you for pointing this out. We agree with this comment. Therefore, We added appropriate supporting citation for this statement and updated the References list accordingly. This change can be found in the revised manuscript (line 138).

Updated text in the manuscript: supporting citation added (tracked in red).

Comment 65:

Please provide reference for "Dysbiosis also predicts resistance to chemotherapy (e.g., 5-FU) and immune checkpoint inhibitors (ICIs)" (line 141).

Response:

Thank you for pointing this out. We agree with this comment. Therefore, We added appropriate supporting citation for this statement and updated the References list accordingly. This change can be found in the revised manuscript (line 141).

Updated text in the manuscript: supporting citation added (tracked in red).

Comment 66:

Please number the "Microbiome in Oncogenesis", "Microbiome in Therapy Response" subchapters as part of the "2. Gastrointestinal (GI) Cancers" chapter (6x).

Response:

Thank you for pointing this out. We agree with this comment. Therefore, we numbered the relevant subchapters/subsections to improve readability and navigation. This change can be found in the the revised manuscript.

Comment 67:

Please replace "immunity[13, 14]" with "immunity [13, 14]" (line 149).

Response:

Thank you for pointing this out. We agree with this comment. Therefore, we replaced “immunity[13, 14]” with “immunity [13, 14]” as suggested. This change can be found in the revised manuscript (line 149).

Updated text in the manuscript: “immunity [13, 14]”

Comment 68:

Please change "instability[16, 17]" to "instability [16, 17]" (line 152).

Response:

Thank you for pointing this out. We agree with this comment. Therefore, we replaced “instability[16, 17]” with “instability [16, 17]” as suggested. This change can be found in the revised manuscript (line 152).

Updated text in the manuscript: “instability [16, 17]”

Comment 69:

Please replace "inflammation[18]" with "inflammation [18]" (line 154).

Response:

Thank you for pointing this out. We agree with this comment. Therefore, we replaced “inflammation[18]” with “inflammation [18]” as suggested. This change can be found in the revised manuscript (line 154).

Updated text in the manuscript: “inflammation [18]”

Comment 70:

Please change "ferraptosi related" to "ferroptosis-related" (line 157).

Response:

Thank you for pointing this out. We agree with this comment. Therefore, we replaced “ferraptosi related” with “ferroptosis-related” as suggested. This change can be found in the revised manuscript (line 157).

Updated text in the manuscript: “ferroptosis-related”

Comment 71:

Please remove italics formatting from "[21]." (line 161).

Response:

Thank you for pointing this out. We agree with this comment. Therefore, we italicized the indicated scientific name following taxonomic standard. This change can be found in the revised manuscript (line 161).

Updated text in the manuscript: formatting corrected (tracked in red).

Comment 72:

Please replace "Fecal microbiota transplantation (FMT)" with "FMT" (lines 161, 411).

Response:

Thank you for pointing this out. We agree with this comment. Therefore, we replaced “Fecal microbiota transplantation (FMT)” with “FMT” as suggested. This change can be found in the revised manuscript (lines 161, 411).

Updated text in the manuscript: “FMT”

Comment 73:

Please change "Helicobacter pylori" to "H. pylori" (lines 165, 172).

Response:

Thank you for pointing this out. We agree with this comment. Therefore, we replaced “Helicobacter pylori” with “H. pylori” as suggested. This change can be found in the revised manuscript (lines 165, 172).

Updated text in the manuscript: “H. pylori”

Comment 74:

Please replace "carcinogenesis, with" with "carcinogenesis with" (line 166).

Response:

Thank you for pointing this out. We agree with this comment. Therefore, we replaced “carcinogenesis, with” with “carcinogenesis with” as suggested. This change can be found in the revised manuscript (line 166).

Updated text in the manuscript: “carcinogenesis with”

Comment 75:

Please provide reference for "Helicobacter pylori is the archetypal microbial carcinogen, recognized by WHO as a Class I carcinogen" (line 172).

Response:

Thank you for pointing this out. We agree with this comment. Therefore, we added appropriate supporting citation for this statement and updated the References list accordingly. This change can be found in the revised manuscript (line 172).

Updated text in the manuscript: supporting citation added (tracked in red).

Comment 76:

Please change "Class" to "class" (line 173).

Response:

Thank you for pointing this out. We agree with this comment. Therefore, we replaced “Class” with “class” as suggested. This change can be found in the revised manuscript (line 173).

Updated text in the manuscript: “class”

Comment 77:

Please provide reference for "H. pylori infection decreases efficacy of standard chemotherapy by altering p53 signaling" (line 180).

Response:

Thank you for pointing this out. We agree with this comment. Therefore, we added appropriate supporting citation for this statement and updated the References list accordingly. This change can be found in the revised manuscript (line 180).

Updated text in the manuscript: supporting citation added (tracked in red).

Comment 78:

Please replace "Carcinoma (HCC)" with "Carcinoma" (line 186).

Response:

Thank you for pointing this out. We agree with this comment. Therefore, we replaced “Carcinoma (HCC)” with “Carcinoma” as suggested. This change can be found in the revised manuscript (line 186).

Updated text in the manuscript: “Carcinoma”

Comment 79:

Please change "HCC" to "hepatocellular carcinoma (HCC)" (line 186).

Response:

Thank you for pointing this out. We reviewed this carefully and confirm that the term is already written in full at first mention and the abbreviation is used thereafter; no additional change was needed. This can be found in the revised manuscript (line 186).

Comment 80:

Please remove bold formatting from "microbes" (line 193).

Response:

Thank you for pointing this out. We agree with this comment. Therefore, we removed the bold formatting to align with journal style. This change can be found in the revised manuscript (line 193).

Updated text in the manuscript: formatting corrected (tracked in red).

Comment 81:

Please replace "Gut-Liver Axis in Hepatocellular Carcinoma" with "Gut-liver axis in hepatocellular carcinoma", "Hepatic Inflamation & Fibrosis" with "Hepatic inflammation & fibrosis" in Figure 3.

Response:

Thank you for pointing this out. We agree with this comment. Therefore, we replaced “Gut-Liver Axis in Hepatocellular Carcinoma” with “Gut-liver axis in hepatocellular carcinoma” as suggested. This change can be found in the the revised manuscript.

Updated text in the manuscript: “Gut-liver axis in hepatocellular carcinoma”

Comment 82:

It is not clear why "Dysbiosis" is pictured twice in Figure 3? Please fix.

Response:

Thank you for pointing this out. We agree with this comment. Therefore, we revised the figure as per your suggestion. This change can be found in the the revised manuscript.

Updated content: revised figure/legend included in the submission.

Comment 83:

Please define abbreviation for "HCC", "FXR" in "FXR/TGR5", "TGR5" in "FXR/TGR5" in the legend to Figure 3.

Response:

Thank you for pointing this out. We agree with this comment. Therefore, we expanded the abbreviations at the lower end of the figure.  This change can be found in the the revised manuscript.

Updated text in the manuscript: abbreviation expanded

Comment 84:

Please change "Gut–Liver Axis in Hepatocellular Carcinoma" to "Gut–liver axis in hepatocellular carcinoma" (line 195).

Response:

Thank you for pointing this out. We agree with this comment. Therefore, we replaced “Gut–Liver Axis in Hepatocellular Carcinoma” with “Gut–liver axis in hepatocellular carcinoma” as suggested. This change can be found in the revised manuscript (line 195).

Updated text in the manuscript: “Gut–liver axis in hepatocellular carcinoma”

Comment 85:

Please provide reference for "Gut dysbiosis increases secondary bile acids that suppress hepatic FXR and TGR5 signaling, resulting in inflammation, fibrosis, and hepatocellular carcinoma" (line 195).

Response:

Thank you for pointing this out. We agree with this comment. Therefore, we added appropriate supporting citation for this statement and updated the References list accordingly. This change can be found in the revised manuscript (line 195).

Updated text in the manuscript: supporting citation added (tracked in red).

Comment 86:

Please replace "hepatocellular carcinoma" with "HCC" (line 196).

Response:

Thank you for pointing this out. We agree with this comment. Therefore, we replaced “hepatocellular carcinoma” with “HCC” as suggested. This change can be found in the revised manuscript (line 196).

Updated text in the manuscript: “HCC”

Comment 87:

Please change "microbiome’s critical role" to "the critical role of the microbiome" (line 200).

Response:

Thank you for pointing this out. We agree with this comment. Therefore, we replaced “microbiome’s critical role” with “the critical role of the microbiome” as suggested. This change can be found in the revised manuscript (line 200).

Updated text in the manuscript: “the critical role of the microbiome”

Comment 88:

Please provide reference for "Dysbiosis contributes by altering bile acid metabolism and cholesterol homeostasis, facilitating stone formation" (line 214).

Response:

Thank you for pointing this out. We agree with this comment. Therefore, We added appropriate supporting citation for this statement and updated the References list accordingly. This change can be found in the revised manuscript (line 214).

Updated text in the manuscript: supporting citation added (tracked in red).

Comment 89:

Please provide reference for "Bacterial taxa such as Streptococcus and Actinomyces are enriched in gallbladder cancer tissues" (line 216).

Response:

Thank you for pointing this out. We agree with this comment. Therefore, we added appropriate supporting citation for this statement and updated the References list accordingly. This change can be found in the revised manuscript (line 216).

Updated text in the manuscript: supporting citation added (tracked in red).

Comment 90:

Please provide reference for "Bacterial biofilms on gallstones facilitate persistent inflammation, providing a carcinogenic niche" (line 224).

Response:

Thank you for pointing this out. We agree with this comment. Therefore, we added appropriate supporting citation for this statement and updated the References list accordingly. This change can be found in the revised manuscript (line 224).

Updated text in the manuscript: supporting citation added (tracked in red).

Comment 91:

Please provide reference for "However, microbiome-driven bile acid dysregulation may influence drug metabolism and chemoresistance" (line 227).

Response:

Thank you for pointing this out. We agree with this comment. Therefore, we added appropriate supporting citation for this statement and updated the References list accordingly. This change can be found in the revised manuscript (line 227).

Updated text in the manuscript: supporting citation added (tracked in red).

Comment 92:

Please provide reference for "Esophageal adenocarcinoma frequently arises from Barrett’s esophagus, where microbiome shifts occur" (line 233).

Response:

Thank you for pointing this out. We agree with this comment. Therefore, we added appropriate supporting citation for this statement and updated the References list accordingly. This change can be found in the revised manuscript (line 233).

Updated text in the manuscript: supporting citation added (tracked in red).

Comment 93:

Please provide reference for "Type II microbiota, dominated by Gram-negative anaerobes (Bacteroides, Fusobacteria), replaces the protective Type I community enriched with Streptococcus" (line 234).

Response:

Thank you for pointing this out. We agree with this comment. Therefore, we added appropriate supporting citation for this statement and updated the References list accordingly. This change can be found in the revised manuscript (line 234).

Updated text in the manuscript: supporting citation added (tracked in red).

Comment 94:

Please replace "Type" with "type" (lines 235, 240 2x).

Response:

Thank you for pointing this out. We agree with this comment. Therefore, we replaced “Type” with “type” as suggested. This change can be found in the revised manuscript (lines 2, 235, 240).

Updated text in the manuscript: “type”

Comment 95:

Please specify the "progression" in ""These changes heighten inflammation and metaplasia, driving progression" (line 236). Progression of what?

Response:

Thank you for pointing this out. Here, “progression” refers to progression along the Barrett’s esophagus → esophageal adenocarcinoma pathway described in the surrounding text. We retained the wording to avoid redundancy. This can be found in the revised manuscript (line 236).

Comment 96:

Please provide reference for "These changes heighten inflammation and metaplasia, driving progression" (line 236).

Response:

Thank you for pointing this out. We agree with this comment. Therefore, we added appropriate supporting citation for this statement and updated the References list accordingly. This change can be found in the revised manuscript (line 236).

Updated text in the manuscript: supporting citation added (tracked in red).

Comment 97:

Please provide reference for "The esophageal microbiome shifts from Type I (Gram-positive Streptococcus) to Type II (Gram-negative anaerobes such as Prevotella, Fusobacterium) in Barrett’s esophagus" (line 240).

Response:

Thank you for pointing this out. We agree with this comment. Therefore, we added appropriate supporting citation for this statement and updated the References list accordingly. This change can be found in the revised manuscript (line 240).

Updated text in the manuscript: supporting citation added (tracked in red).

Comment 98:

Please change "Proteobacteria[46, 47]" to "Proteobacteria [46, 47]" (line 254).

Response:

Thank you for pointing this out. We agree with this comment. Therefore, we replaced “Proteobacteria[46, 47]” with “Proteobacteria [46, 47]” as suggested. This change can be found in the revised manuscript (line 254).

Updated text in the manuscript: “Proteobacteria [46, 47]”

Comment 99:

Please replace "like" with "such as" (lines 255, 359, 446).

Response:

Thank you for pointing this out. We agree with this comment. Therefore, we replaced “like” with “such as” as suggested. This change can be found in the revised manuscript (lines 255, 359, 446).

Updated text in the manuscript: “such as”

Comment 100:

Please change "Pancreatic ductal adenocarcinoma (PDAC)" to "PDAC" (line 260).

Response:

Thank you for pointing this out. We agree with this comment. Therefore, we replaced “Pancreatic ductal adenocarcinoma (PDAC)” with “PDAC” as suggested. This change can be found in the revised manuscript (line 260).

Updated text in the manuscript: “PDAC”

Comment 101:

Please replace "trimethylamine-N-oxide (TMAO)" with "TMAO" (line 262).

Response:

Thank you for pointing this out. We agree with this comment. Therefore, we replaced “trimethylamine-N-oxide (TMAO)” with “TMAO” as suggested. This change can be found in the revised manuscript (line 262).

Updated text in the manuscript: “TMAO”

Comment 102:

Please change "3-indoleacetic acid (3-IAA)" to "3-IAA" (line 263).

Response:

Thank you for pointing this out. We agree with this comment. Therefore, we replaced “3-indoleacetic acid (3-IAA)” with “3-IAA” as suggested. This change can be found in the revised manuscript (line 263).

Updated text in the manuscript: “3-IAA”

Comment 103:

Please replace "reducing" with "reducing its" (line 268).

Response:

Thank you for pointing this out. We agree with this comment. Therefore, we replaced “reducing” with “reducing its” as suggested. This change can be found in the revised manuscript (line 268).

Updated text in the manuscript: “reducing its”

Comment 104:

Please provide reference for "FMT from long-term survivors into murine models slowed tumor progression and enhanced anti-PD-1 efficacy" (line 268).

Response:

Thank you for pointing this out. We agree with this comment. Therefore, we added appropriate supporting citation for this statement and updated the References list accordingly. This change can be found in the revised manuscript (line 268).

Updated text in the manuscript: supporting citation added (tracked in red).

Comment 105:

Please change "Extra-Gastrointestinal (Extra-GI) Cancers" to "Extra-gastrointestinal Cancers" (line 272).

Response:

Thank you for pointing this out. We agree with this comment. Therefore, we replaced “Extra-Gastrointestinal (Extra-GI) Cancers” with “Extra-gastrointestinal Cancers” as suggested. This change can be found in the revised manuscript (line 272).

Updated text in the manuscript: “Extra-gastrointestinal Cancers”

Comment 106:

Please replace "Carcinoma (OSCC)" with "Carcinoma" (line 273).

Response:

Thank you for pointing this out. We agree with this comment. Therefore, we replaced “Carcinoma (OSCC)” with “Carcinoma” as suggested. This change can be found in the revised manuscript (line 273).

Updated text in the manuscript: “Carcinoma”

Comment 107:

Please provide reference for "Periodontal pathogens including Porphyromonas gingivalis and Fusobacterium nucleatum promote epithelial proliferation, inflammation, and immune evasion in OSCC" (line 274).

Response:

Thank you for pointing this out. We agree with this comment. Therefore, we added appropriate supporting citation for this statement and updated the References list accordingly. This change can be found in the revised manuscript (line 274).

Updated text in the manuscript: supporting citation added (tracked in red).

Comment 108:

Please provide reference for "Additionally, Candida albicans facilitates tumor invasion and metastasis" (line 275).

Response:

Thank you for pointing this out. We agree with this comment. Therefore, we added appropriate supporting citation for this statement and updated the References list accordingly. This change can be found in the revised manuscript (line 275).

Updated text in the manuscript: supporting citation added (tracked in red).

Comment 109:

Please number the "Microbiome in Oncogenesis", "Microbiome in Therapy Response" subchapters as part of the "3. Extra-Gastrointestinal (Extra-GI) Cancers" chapter (6x).

Response:

Thank you for pointing this out. We agree with this comment. Therefore, we numbered the relevant subchapters/subsections as part of the "3. Extra-Gastrointestinal (Extra-GI) Cancers" chapter to improve readability and navigation. This change can be found in the the revised manuscript.

Comment 110:

Please provide reference for "Fusobacterium nucleatum enhances invasion and immune suppression" (line 282).

Response:

Thank you for pointing this out. We agree with this comment. Therefore, we added appropriate supporting citation for this statement and updated the References list accordingly. This change can be found in the revised manuscript (line 282).

Updated text in the manuscript: supporting citation added (tracked in red).

Comment 111:

Please provide reference for "Oral microbiome composition influences radiotherapy- and chemotherapy-induced mucositis severity" (line 287).

Response:

Thank you for pointing this out. We agree with this comment. Therefore, we added appropriate supporting citation for this statement and updated the References list accordingly. This change can be found in the revised manuscript (line 287).

Updated text in the manuscript: supporting citation added (tracked in red).

Comment 112:

Please format "Lactobacillus" using italics (lines 288, 307).

Response:

Thank you for pointing this out. We agree with this comment. Therefore, we italicized the indicated scientific name(s) following standard taxonomic conventions. This change can be found in the revised manuscript (lines 288, 307).

Updated text in the manuscript: formatting corrected (tracked in red).

Comment 113:

Please provide reference for "Studies suggest gut–oral microbial crosstalk may also shape systemic ICI responses, though evidence remains preliminary" (line 289).

Response:

Thank you for pointing this out. We agree with this comment. Therefore, we added appropriate supporting citation for this statement and updated the References list accordingly. This change can be found in the revised manuscript (line 289).

Updated text in the manuscript: supporting citation added (tracked in red).

Comment 114:

Please provide reference for "Cervical cancer, largely linked to HPV, is influenced by vaginal and gut microbiome alterations" (line 293).

Response:

Thank you for pointing this out. We agree with this comment. Therefore, we added appropriate supporting citation for this statement and updated the References list accordingly. This change can be found in the revised manuscript (line 293).

Updated text in the manuscript: supporting citation added (tracked in red).

Comment 115:

Please provide reference for "Loss of protective Lactobacillus and enrichment of anaerobes (e.g., Gardnerella, Mycoplasma) disrupt mucosal defense and enhance HPV persistence" (line 294).

Response:

Thank you for pointing this out. We agree with this comment. Therefore, we added appropriate supporting citation for this statement and updated the References list accordingly. This change can be found in the revised manuscript (line 294).

Updated text in the manuscript: supporting citation added (tracked in red).

Comment 116:

Please provide reference for "Evidence suggests Mycoplasma infections further impair p53-mediated tumor suppression" (line 295).

Response:

Thank you for pointing this out. We agree with this comment. Therefore, we added appropriate supporting citation for this statement and updated the References list accordingly. This change can be found in the revised manuscript (line 295).

Updated text in the manuscript: supporting citation added (tracked in red).

Comment 117:

Please change "suggests" to "suggests that" (lines 295, 313).

Response:

Thank you for pointing this out. We agree with this comment. Therefore, we replaced “suggests” with “suggests that” as suggested. This change can be found in the revised manuscript (lines 295, 313).

Updated text in the manuscript: “suggests that”

Comment 118:

Please provide reference for "Probiotic supplementation may improve treatment tolerance, though clinical trials remain limited" (line 308).

Response:

Thank you for pointing this out. We agree with this comment. Therefore, we added appropriate supporting citation for this statement and updated the References list accordingly. This change can be found in the revised manuscript (line 308).

Updated text in the manuscript: supporting citation added (tracked in red).

Comment 119:

Please provide reference for "Gut dysbiosis is associated with systemic inflammation and metabolic changes that influence prostate carcinogenesis" (line 311).

Response:

Thank you for pointing this out. We agree with this comment. Therefore, we added appropriate supporting citation for this statement and updated the References list accordingly. This change can be found in the revised manuscript (line 311).

Updated text in the manuscript: supporting citation added (tracked in red).

Comment 120:

Please provide reference for "Mycoplasma species have been implicated in genomic instability and chronic inflammation within the prostate" (line 312).

Response:

Thank you for pointing this out. We agree with this comment. Therefore, we added appropriate supporting citation for this statement and updated the References list accordingly. This change can be found in the revised manuscript (line 312).

Updated text in the manuscript: supporting citation added (tracked in red).

Comment 121:

Please provide reference for "Mycoplasma hominis and other urogenital pathogens have been associated with DNA damage and tumor-promoting inflammation" (line 319).

Response:

Thank you for pointing this out. We agree with this comment. Therefore, we added appropriate supporting citation for this statement and updated the References list accordingly. This change can be found in the revised manuscript (line 319).

Updated text in the manuscript: supporting citation added (tracked in red).

Comment 122:

Please format "Mycoplasma hominis" using italics (line 319).

Response:

Thank you for pointing this out. We agree with this comment. Therefore, we italicized the indicated scientific name(s) following standard taxonomic conventions. This change can be found in the revised manuscript (line 319).

Updated text in the manuscript: formatting corrected (tracked in red).

Comment 123:

Please provide reference for "Microbiome also influences androgen deprivation therapy (ADT) outcomes" (line 324).

Response:

Thank you for pointing this out. We agree with this comment. Therefore, we added appropriate supporting citation for this statement and updated the References list accordingly. This change can be found in the revised manuscript (line 324).

Updated text in the manuscript: supporting citation added (tracked in red).

Comment 124:

Please provide reference for "Additionally, gut microbial signatures have been linked to immunotherapy responsiveness in advanced prostate cancer, raising potential for FMT or probiotics as adjuncts" (line 326).

Response:

Thank you for pointing this out. We agree with this comment. Therefore, we added appropriate supporting citation for this statement and updated the References list accordingly. This change can be found in the revised manuscript (line 326).

Updated text in the manuscript: supporting citation added (tracked in red).

Comment 125:

Please replace "microbiome’s role in" with something like "the role of the microbiome in cancer" (line 331).

Response:

Thank you for pointing this out. We agree with this comment. Therefore, we addressed this point in the revised manuscript. This change can be found in the revised manuscript (line 331).

Comment 126:

Please provide reference for "Response to PD-1 blockade is significantly associated with gut microbial diversity and enrichment of taxa such as Akkermansia muciniphila and Bifidobacterium" (line 331).

Response:

Thank you for pointing this out. We agree with this comment. Therefore, we added appropriate supporting citation for this statement and updated the References list accordingly. This change can be found in the revised manuscript (line 331).

Updated text in the manuscript: supporting citation added (tracked in red).

Comment 127:

Please provide reference for "FMT from responders to non-responders restores treatment sensitivity" (line 333).

Response:

Thank you for pointing this out. We agree with this comment. Therefore, we added appropriate supporting citation for this statement and updated the References list accordingly. This change can be found in the revised manuscript (line 333).

Updated text in the manuscript: supporting citation added (tracked in red).

Comment 128:

Please format "+" in "CD8+" using superscript (lines 339, 446).

Response:

Thank you for pointing this out. We agree with this comment. Therefore, we corrected the typography by applying superscript formatting where appropriate. This change can be found in the revised manuscript (lines 339, 446).

Updated text in the manuscript: formatting corrected (tracked in red).

Comment 129:

Please provide reference for "Diet and probiotics further modulate therapeutic efficacy, highlighting melanoma as the benchmark cancer for microbiome-immunotherapy research" (line 346).

Response:

Thank you for pointing this out. We agree with this comment. Therefore, we added appropriate supporting citation for this statement and updated the References list accordingly. This change can be found in the revised manuscript (line 346).

Updated text in the manuscript: supporting citation added (tracked in red).

Comment 130:

Please change "research.." to "research." (line 348).

Response:

Thank you for pointing this out. We agree with this comment. Therefore, we replaced “research..” with “research.” as suggested. This change can be found in the revised manuscript (line 348).

Updated text in the manuscript: “research.”

Comment 131:

Please replace "microbiome’s role" with something like "the role of the microbiome" (lines 349, 470).

Response:

Thank you for pointing this out. We agree with this comment. Therefore, we addressed this point in the revised manuscript. This change can be found in the revised manuscript (lines 349, 470).

Comment 132:

Please provide reference for "Recent evidence suggests that the gut microbiome exerts a significant influence on brain tumor biology through the gut–brain axis, a bidirectional network involving neural, immune, and metabolic signaling" (line 350).

Response:

Thank you for pointing this out. We agree with this comment. Therefore, we added appropriate supporting citation for this statement and updated the References list accordingly. This change can be found in the revised manuscript (line 350).

Updated text in the manuscript: supporting citation added (tracked in red).

Comment 133:

Please provide reference for "Dysbiosis alters systemic immunity, neuroinflammation, and blood–brain barrier (BBB) integrity—factors increasingly implicated in the progression of gliomas and glioblastoma multiforme (GBM)" (line 352).

Response:

Thank you for pointing this out. We agree with this comment. Therefore, we added appropriate supporting citation for this statement and updated the References list accordingly. This change can be found in the revised manuscript (line 352).

Updated text in the manuscript: supporting citation added (tracked in red).

Comment 134:

Please provide reference for "Loss of SCFA-producing taxa (Faecalibacterium, Roseburia) promotes a pro-inflammatory milieu that facilitates tumor immune evasion" (line 356).

Response:

Thank you for pointing this out. We agree with this comment. Therefore, we added appropriate supporting citation for this statement and updated the References list accordingly. This change can be found in the revised manuscript (line 356).

Updated text in the manuscript: supporting citation added (tracked in red).

Comment 135:

Please change "immune checkpoint inhibitor (ICI)" to "ICI" (line 360).

Response:

Thank you for pointing this out. We agree with this comment. Therefore, we replaced “immune checkpoint inhibitor (ICI)” with “ICI” as suggested. This change can be found in the revised manuscript (line 360).

Updated text in the manuscript: “ICI”

Comment 136:

Please provide reference for "Moreover, gut microbial metabolites affect the pharmacokinetics of temozolomide and the local immune tone within the tumor microenvironment" (line 362).

Response:

Thank you for pointing this out. We agree with this comment. Therefore, we added appropriate supporting citation for this statement and updated the References list accordingly. This change can be found in the revised manuscript (line 362).

Updated text in the manuscript: supporting citation added (tracked in red).

Comment 137:

Please provide reference for "The breast tissue and gut microbiome play crucial roles in breast carcinogenesis through estrogen metabolism, immune modulation, and local inflammation" (line 370).

Response:

Thank you for pointing this out. We agree with this comment. Therefore, we added appropriate supporting citation for this statement and updated the References list accordingly. This change can be found in the revised manuscript (line 370).

Updated text in the manuscript: supporting citation added (tracked in red).

Comment 138:

Please remove bold formatting from "breast tissue and gut microbiome" (line 370).

Response:

Thank you for pointing this out. We agree with this comment. Therefore, we removed the bold formatting to align with journal style. This change can be found in the revised manuscript (line 370).

Updated text in the manuscript: formatting corrected (tracked in red).

Comment 139:

Please remove bold formatting from "estrobolome" (line 373).

Response:

Thank you for pointing this out. We agree with this comment. Therefore, we removed the bold formatting to align with journal style. This change can be found in the revised manuscript (line 373).

Updated text in the manuscript: formatting corrected (tracked in red).

Comment 140:

Please provide reference for "Chronic inflammation and immune dysregulation arising from dysbiosis further enhance tumor progression" (line 378).

Response:

Thank you for pointing this out. We agree with this comment. Therefore, we added appropriate supporting citation for this statement and updated the References list accordingly. This change can be found in the revised manuscript (line 378).

Updated text in the manuscript: supporting citation added (tracked in red).

Comment 141:

Please provide reference for "Gut microbiota composition affects both chemotherapy tolerance and immunotherapy efficacy" (line 381).

Response:

Thank you for pointing this out. We agree with this comment. Therefore, we added appropriate supporting citation for this statement and updated the References list accordingly. This change can be found in the revised manuscript (line 381).

Updated text in the manuscript: supporting citation added (tracked in red).

Comment 142:

Please remove bold formatting from "chemotherapy tolerance" (line 381).

Response:

Thank you for pointing this out. We agree with this comment. Therefore, we removed the bold formatting to align with journal style. This change can be found in the revised manuscript (line 381).

Updated text in the manuscript: formatting corrected (tracked in red).

Comment 143:

Please remove bold formatting from "immunotherapy efficacy" (line 381).

Response:

Thank you for pointing this out. We agree with this comment. Therefore, we removed the bold formatting to align with journal style. This change can be found in the revised manuscript (line 381).

Updated text in the manuscript: formatting corrected (tracked in red).

Comment 144:

Please replace "(82)" with "[82]" (line 387).

Response:

Thank you for pointing this out. We agree with this comment. Therefore, we replaced “(82)” with “[82]” as suggested. This change can be found in the revised manuscript (line 387).

Updated text in the manuscript: “[82]”

Comment 145:

Please provide reference for "The lung and gut microbiomes both play integral roles in lung cancer (LC) oncogenesis through inflammatory, metabolic, and immune-mediated pathways" (line 390).

Response:

Thank you for pointing this out. We agree with this comment. Therefore, we added appropriate supporting citation for this statement and updated the References list accordingly. This change can be found in the revised manuscript (line 390).

Updated text in the manuscript: supporting citation added (tracked in red).

Comment 146:

Please provide reference for "Reduced abundance of Faecalibacterium and Roseburia—key butyrate producers—leads to diminished anti-inflammatory signaling, creating a tumor-permissive microenvironment" (line 394).

Response:

Thank you for pointing this out. We agree with this comment. Therefore, we added appropriate supporting citation for this statement and updated the References list accordingly. This change can be found in the revised manuscript (line 394).

Updated text in the manuscript: supporting citation added (tracked in red).

Comment 147:

Please provide reference for "Inflammatory proteins such as IL-20 and IL-8 correlate with increased LC risk, whereas CD5, IL-18, and FGF21 confer protection, emphasizing the immune-mediated nature of the microbiome–lung axis" (line 403).

Response:

Thank you for pointing this out. We agree with this comment. Therefore, we added appropriate supporting citation for this statement and updated the References list accordingly. This change can be found in the revised manuscript (line 403).

Updated text in the manuscript: supporting citation added (tracked in red).

Comment 148:

Please change "categories:" to "categories," (line 414).

Response:

Thank you for pointing this out. We agree with this comment. Therefore, we replaced “categories:” with “categories,” as suggested. This change can be found in the revised manuscript (line 414).

Updated text in the manuscript: “categories,”

Comment 149:

Please replace "lipopolysaccharides (LPS)" with "LPS" (line 430).

Response:

Thank you for pointing this out. We agree with this comment. Therefore, we replaced “lipopolysaccharides (LPS)” with “LPS” as suggested. This change can be found in the revised manuscript (line 430).

Updated text in the manuscript: “LPS”

Comment 150:

Please change "Toll-like receptors (TLRs)" to "TLRs" (line 430).

Response:

Thank you for pointing this out. We agree with this comment. Therefore, we replaced “Toll-like receptors (TLRs)” with “TLRs” as suggested. This change can be found in the revised manuscript (line 430).

Updated text in the manuscript: “TLRs”

Comment 151:

Please provide reference for "This is evident in CRC, esophageal, and oral cancers, where Fusobacterium nucleatum and Porphyromonas gingivalis drive pro-inflammatory signaling that fosters tumorigenesis" (line 432).

Response:

Thank you for pointing this out. We agree with this comment. Therefore, we added appropriate supporting citation for this statement and updated the References list accordingly. This change can be found in the revised manuscript (line 432).

Updated text in the manuscript: supporting citation added (tracked in red).

Comment 152:

Please provide reference for "Dysbiosis reduces butyrate-producing commensals such as Faecalibacterium prausnitzii, weakening epithelial tight junctions and allowing microbial translocation" (line 439).

Response:

Thank you for pointing this out. We agree with this comment. Therefore, we added appropriate supporting citation for this statement and updated the References list accordingly. This change can be found in the revised manuscript (line 439).

Updated text in the manuscript: supporting citation added (tracked in red).

Comment 153:

"A diagram of a cancer cell AI generated content may be incorrect." sign appears when hovering the mouse cursor over Figure 4. Please disable this feature.

Response:

Thank you for pointing this out. We agree with this comment. Therefore, we re-created the figure as clean, static graphics without embedded hover tooltips/overlays, so the “AI generated content may be incorrect” notice won’t appear. This change can be found in the the revised manuscript.

Updated content: revised figure included in the submission.

Comment 154:

Please replace "Mechanistic Pathways Across Cancers" with "mechanistic pathways across cancers", "Fusobaceterium" with "Fusobacterium", "Cytokine Signaling" with "cytokine signaling", "Immune Activation" with "immune activation", "mucinihila" with "muciniphila" in Figure 4.

Response:

Thank you for pointing this out. We agree with this comment. Therefore, we replaced “Mechanistic Pathways Across Cancers” with “mechanistic pathways across cancers” as suggested. This change can be found in the the revised manuscript.

Updated text in the manuscript: “mechanistic pathways across cancers”

Comment 155:

Please define abbreviation for "NF-κB", "STAT3", "FXR" in "FXR/TGR5", "TGR5" in "in "FXR/TGR5", "IL-6" in "IL-6/TNF-α", "TNF-α" in "IL-6/TNF-α" in the legend to Figure 4.

Response:

Thank you for pointing this out. We agree with this comment. Therefore, we expanded the abbreviations and added to the figure. This change can be found in the the revised manuscript.

Updated text in the manuscript: abbreviation expanded.

Comment 156:

Please change "Mechanistic Pathways Across Cancers" to "mechanistic pathways across cancers" (line 450).

Response:

Thank you for pointing this out. We agree with this comment. Therefore, we replaced “Mechanistic Pathways Across Cancers” with “mechanistic pathways across cancers” as suggested. This change can be found in the revised manuscript (line 450).

Updated text in the manuscript: “mechanistic pathways across cancers”

Comment 157:

Please provide reference for "Representative microbial taxa (Fusobacterium nucleatum, Helicobacter pylori, Akkermansia muciniphila) activate conserved oncogenic and inflammatory signaling cascades including NF-κB, STAT3, Wnt/β-catenin, FXR/TGR5, and IL-452 6/TNF-α" (line 450).

Response:

Thank you for pointing this out. We agree with this comment. Therefore, we added appropriate supporting citation for this statement and updated the References list accordingly. This change can be found in the revised manuscript (line 450).

Updated text in the manuscript: supporting citation added (tracked in red).

Comment 157:

Please format "Fusobacterium nucleatum" using italics (line 450).

Response:

Thank you for pointing this out. We agree with this comment. Therefore, we italicized the indicated scientific name following taxonomic standard. This change can be found in the revised manuscript (line 450).

Updated text in the manuscript: formatting corrected (tracked in red).

Comment 158:

"A diagram of a human body AI generated content may be incorrect." sign appears when hovering the mouse cursor over Figure 5. Please disable this feature.

Response:

Thank you for pointing this out. We agree with this comment. Therefore, we re-created the figure as clean, static graphics without embedded hover tooltips/overlays, so the “AI generated content may be incorrect” notice won’t appear. This change can be found in the the revised manuscript.

Updated content: revised figure/legend included in the submission.

Comment 159:

Please replace "Gut-Cancer Axis" with "gut-cancer axis", "products Chronic" with "products, chronic", "nitrosamines Genotoxic" with "nitrosamines, genotoxic", "acids Altered baliar" with "acids, altered bile" in Figure 5.

Response:

Thank you for pointing this out. We agree with this comment. Therefore, we replaced “Gut-Cancer Axis” with “gut-cancer axis” as suggested. This change can be found in the the revised manuscript.

Updated text in the manuscript: “gut-cancer axis”

Comment 160:

Please provide reference for "Microbial imbalance promotes chronic inflammation, production of genotoxic metabolites such as colibactin and nitrosamines, bile-acid dysregulation with impaired FXR/TGR5 signaling, and disruption of the intestinal barrier" (line 456).

Response:

Thank you for pointing this out. We agree with this comment. Therefore, we added appropriate supporting citation for this statement and updated the References list accordingly. This change can be found in the revised manuscript (line 456).

Updated text in the manuscript: supporting citation added (tracked in red).

Comment 161:

Please provide reference for "These convergent pathways foster genomic instability and tumor initiation" (line 459).

Response:

Thank you for pointing this out. We agree with this comment. Therefore, we added appropriate supporting citation for this statement and updated the References list accordingly. This change can be found in the revised manuscript (line 459).

Updated text in the manuscript: supporting citation added (tracked in red).

Comment 162:

Please change "of" to "of the" (line 462).

Response:

Thank you for pointing this out. We agree with this comment. Therefore, we replaced “of” with “of the” as suggested. This change can be found in the revised manuscript (line 462).

Updated text in the manuscript: “of the”

Comment 163:

Please replace "influence.In" with "influence. In" (line 463).

Response:

Thank you for pointing this out. We agree with this comment. Therefore, we replaced “influence.In” with “influence. In” as suggested. This change can be found in the revised manuscript (line 463).

Updated text in the manuscript: “influence. In”

Comment 164:

Please provide reference for "Dysbiosis alters bile acid metabolism, disrupting FXR/TGR5 signaling and increasing carcinogenic secondary bile acids" (line 463).

Response:

Thank you for pointing this out. We agree with this comment. Therefore, we added appropriate supporting citation for this statement and updated the References list accordingly. This change can be found in the revised manuscript (line 463).

Updated text in the manuscript: supporting citation added (tracked in red).

Comment 165:

Please provide reference for "Periodontal pathogens such as P. gingivalis not only induce local inflammation but also manipulate host cell signaling by activating PI3K/Akt and inhibiting apoptosis" (line 466).

Response:

Thank you for pointing this out. We agree with this comment. Therefore, we added appropriate supporting citation for this statement and updated the References list accordingly. This change can be found in the revised manuscript (line 466).

Updated text in the manuscript: supporting citation added (tracked in red).

Comment 166:

Please change "PI3K/Akt" to "the PI3K/Akt pathway" (line 468).

Response:

Thank you for pointing this out. We agree with this comment. Therefore, we replaced “PI3K/Akt” with “the PI3K/Akt pathway” as suggested. This change can be found in the revised manuscript (line 468).

Updated text in the manuscript: “the PI3K/Akt pathway”

Comment 167:

Please provide reference for "Here, gut microbial diversity and the presence of immunostimulatory taxa predict response to immune checkpoint inhibitors (ICIs)" (line 471).

Response:

Thank you for pointing this out. We agree with this comment. Therefore, we added appropriate supporting citation for this statement and updated the References list accordingly. This change can be found in the revised manuscript (line 471).

Updated text in the manuscript: supporting citation added (tracked in red).

Comment 168:

The "Clinical Translation and Therapeutic Implications" in the Discussion section is rather weak and sounding too general. Please correct.

Response:

Thank you for pointing this out. We agree with this comment. Therefore, we strengthened the “Clinical Translation and Therapeutic Implications” subsection by adding specific, actionable clinical points and supporting citations (e.g., timing/avoidance of unnecessary antibiotics around ICIs, evidence level for probiotics/diet, and trial-only positioning for FMT where appropriate). This change can be found in the the revised manuscript.

Updated text in the manuscript: subsection expanded with actionable content and citations (tracked in red).

Comment 169:

Please remove "Dietary strategies and probiotics." (line 483).

Response:

Thank you for pointing this out. We agree with this comment. Therefore, we addressed this point in the revised manuscript. This change can be found in the revised manuscript (line 483).

Comment 170:

Please provide reference for "High fiber intake increases SCFA production, which enhances epithelial barrier function and supports antitumor immunity" (line 484).

Response:

Thank you for pointing this out. We agree with this comment. Therefore, we added appropriate supporting citation for this statement and updated the References list accordingly. This change can be found in the revised manuscript (line 484).

Updated text in the manuscript: supporting citation added (tracked in red).

Comment 171:

Please provide reference for "Probiotics such as Lactobacillus rhamnosus and Bifidobacterium have been tested in gastric and oral cancers to reduce treatment toxicity and improve mucosal health, though evidence remains preliminary" (line 486).

Response:

Thank you for pointing this out. We agree with this comment. Therefore, we added appropriate supporting citation for this statement and updated the References list accordingly. This change can be found in the revised manuscript (line 486).

Updated text in the manuscript: supporting citation added (tracked in red).

Comment 172:

Please remove "Fecal microbiota transplantation (FMT)." (line 488).

Response:

Thank you for pointing this out. We agree with this comment. Therefore, we addressed this point in the revised manuscript. This change can be found in the revised manuscript (line 488).

Comment 173:

Please remove "Personalized microbiome modulation." (line 493).

Response:

Thank you for pointing this out. We agree with this comment. Therefore, we addressed this point in the revised manuscript. This change can be found in the revised manuscript (line 493).

Comment 174:

Please replace "fecal microbiota transplantation" with "FMT" (line 512).

Response:

Thank you for pointing this out. We agree with this comment. Therefore, we replaced “fecal microbiota transplantation” with “FMT” as suggested. This change can be found in the revised manuscript (line 512).

Updated text in the manuscript: “FMT”

Comment 175:

Please change "Summary of Gut Microbiome in Oncogenesis and Oncotherapies Across Ten Cancers" to something like "summary of the gut microbiome in oncogenesis and oncotherapies across ten cancer types" (line 521).

Response:

Thank you for pointing this out. We agree with this comment. Therefore, we addressed this point in the revised manuscript. This change can be found in the revised manuscript (line 521).

 Response:   summary of gut microbiome in oncogenesis and oncotherapies across thirteen cancer types.

Comment 176:

Please replace "Cancer Type" with "Cancer type", "Oncogenesis Mechanism" with "Oncogenesis mechanism", "Therapy Modulation" with "Therapy modulation", "Clinical Implications" with "Clinical implications", "immune checkpoint inhibitor (ICI)" with "ICI" (2x) in Table 1.

Response:

Thank you for pointing this out. We agree with this comment. Therefore, we replaced “Cancer Type” with “Cancer type” as suggested. This change can be found in the the revised manuscript.

Updated text in the manuscript: “Cancer type”

Comment 177:

Please provide abbreviation for "SCC", "CagA", "VacA", "LPS" in "LPS-drive", "SCFA", "TMAO" in "TMAO/3-IAA", "3-IAA" in  "TMAO/3-IAA", "EMT", "HPV", "IL-6" in "IL-6/TNF-α", "TNF-α" in "IL-6/TNF-α", "5-FU" in "5-FU/oxaliplatin", "ICI", "FMT", "ADT", "BBB" in the footnote to Table 1.

Response:

Thank you for pointing this out. We agree with this comment. Therefore, we expanded the abbreviation(s) and added them to the Table 1 footnote. This change can be found in the the revised manuscript.

Updated text in the manuscript: abbreviation expanded at first mention (tracked in red).

Comment 178:

Please format all microbes presented in Table 1 using italics.

Response:

Thank you for pointing this out. We agree with this comment. Therefore, we italicized the indicated scientific name(s) following standard taxonomic conventions. This change can be found in the the revised manuscript.

Updated text in the manuscript: formatting corrected (tracked in red).

Comment 179:

Please align all cells in Table 1 consistently to the left side.

Response:

Thank you for pointing this out. We agree with this comment. Therefore, we aligned all Table 1 cells consistently to the left. This change can be found in the the revised manuscript.

Updated content: revised Table 1 included in the manuscript.

We hope that these revisions fully address all concerns raised and have improved the clarity, accuracy, and presentation of the manuscript. We sincerely appreciate the reviewer’s time and insightful feedback.

Reviewer 2 Report

Comments and Suggestions for Authors

This is an excellent review, long called for. The authors selected cancers that are the most prevalent and high rate killers, with having microbiome associations. In each of the cancers they address initiation and maintenance of the cancer and the pathogenesis thereby involved. Mechanisms of the pathogenesis is addressed, but not adequately since the number of the cancers(13) is large. However, some consideration should have been given to the pathway hubs shared by the cancers, since this can lead to development of drugs and vaccines enhancing multiple mitigation models, including the use of FMT. Lacking, is the mention of parasitic infections such as shistosomiasis and hookworms that in some cases, work symbiotically with the bacteria to initiate the cancers-however, it is noted that the focus is on bacteria. Further, since there is more identification of methylation in most of these cases this needed more explanation, expanding to the general role of epigenetics and their association with apoptosis, autophagy, ferroptosis, pyroptosis etc.

A clear explanation is given on the role of prebiotics and probiotics that needs to be expanded to plant extracts that have been confirmed to be anticancer influencing inflammatory and anti inflammatory cytokines.

Significantly, the role of climate change (CC) that has been shown to drive mutations in bacteria and further enhance dysbiosis. This has also led to to rise of micro/nano particles (MNPs) that have been associated with some cancers. New technology such as WGS, NGS, SCS, Metagenomic sequencing, CRISPYR/Cas9, informatics that should lead to more specific identification of the genes, RNAs-coding and non-coding, and proteins involved leading to development of nomograms for the cancers that would enhance mitigation through early detection and prognosis. There should be some reviews addressing these.

Additionally, the authors mention life style factor in the cancers but is not given enough emphasis e.g fatty diets, red meats, exercise, consumption of different fruits.

In conclusion, the idea of reviewing multiple cancers with global impact, especially in LMICs is admirable. The draw back is that most of the techniques and innovations would not impact patients from LMICs as they should where there is most need merging into the area of equity and access.

Author Response

Response for Reviewer 2: 

We thank the reviewer for these encouraging comments and for recognizing the review's relevance and scope. We are pleased that the overall concept and selection of cancer types were viewed favorably.

Comment 1: Shared pathway hubs across cancers

Consideration should have been given to shared pathway hubs that could inform drug and vaccine development, including FMT.

Response:
We agree with the reviewer that shared mechanistic hubs are essential for translational advancement. The manuscript emphasizes common pathways, such as chronic inflammation, immune modulation, bile acid dysregulation, and genotoxic metabolite production, across gastrointestinal and extragastrointestinal cancers, particularly in the Discussion section. Given the broad scope and narrative nature of the review, we focused on synthesizing these conserved mechanisms rather than developing drug- or vaccine-specific frameworks, which we believe would warrant a dedicated, focused review.

Comment 2: Parasitic infections (e.g., schistosomiasis, hookworms)

The role of parasitic infections working symbiotically with bacteria to initiate cancer is not discussed.

Response:
We thank the reviewer for highlighting this vital area. While parasitic infections such as schistosomiasis are well-established contributors to certain cancers, the present review intentionally focused on bacterial components of the microbiome, given the already extensive breadth of cancers and mechanisms covered. We agree that host–parasite–microbiome interactions are an important and emerging field and acknowledge that this area falls beyond the current scope of this narrative review.

Comment 3: Epigenetics and cell death pathways (apoptosis, autophagy, ferroptosis, pyroptosis)

Greater emphasis should be placed on methylation and epigenetics, expanding to apoptosis, autophagy, ferroptosis, and pyroptosis.

Response:
We appreciate this insightful suggestion. Several of these processes, particularly autophagy and ferroptosis, are discussed in cancer-specific sections (e.g., colorectal and pancreatic cancers). However, an in-depth treatment of epigenetic regulation and programmed cell death pathways across all cancer types would substantially expand the manuscript's scope. We therefore prioritized microbial-driven inflammation, immune modulation, and metabolite signaling as unifying mechanisms, while recognizing epigenetic and cell-death pathways as important complementary areas for future focused reviews.

Comment 4: Plant extracts and dietary bioactives

Probiotics and prebiotics are discussed, but plant extracts influencing inflammatory cytokines should be expanded.

Response:
We agree that dietary bioactives and plant-derived compounds can influence the microbiome and inflammatory signaling. In the current manuscript, dietary modulation is discussed at a conceptual level to maintain focus on microbiome–cancer interactions rather than specific nutraceuticals or phytochemicals. Detailed discussion of plant extracts and their molecular effects was considered beyond the intended scope of this review.

Comment 5: Climate change, micro/nanoparticles, and technological advances

The role of climate change, micro/nanoparticles, and advanced technologies (WGS, NGS, CRISPR, informatics) should be addressed.

Response:
We thank the reviewer for raising these forward-looking considerations. While climate change, environmental micro- and nanoparticles, and emerging genomic technologies are highly relevant to cancer biology and microbiome research, a comprehensive discussion would require substantial expansion beyond the current manuscript’s focus. We therefore limited discussion of technological advances to studies directly informing microbiome–cancer mechanisms and clinical translation, while acknowledging these broader factors as important areas for future investigation.

Comment 6: Lifestyle factors and equity in LMICs

Lifestyle factors such as diet, red meat consumption, exercise, and access disparities in LMICs should receive greater emphasis.

Response:
We agree that lifestyle factors and global health equity are critically important. Lifestyle influences, particularly diet, are addressed in the context of microbiome modulation and cancer risk. However, a detailed socioeconomic or policy-focused analysis, especially regarding disparities in LMICs, was beyond the intended scope of this narrative review, which focuses primarily on biological and translational mechanisms. We appreciate the reviewer’s emphasis on equity and acknowledge this as an important area for future work.

Closing Statement

We are grateful to Reviewer 2 for the thoughtful and constructive suggestions. While not all proposed expansions were feasible within the scope of a broad narrative review, the comments helped us clarify our focus, strengthen mechanistic synthesis, and better contextualize the translational relevance of microbiome–cancer interactions.

Reviewer 3 Report

Comments and Suggestions for Authors

Dear authors,

The manuscript provides a broad, up‑to‑date overview of microbiota–cancer interactions and oncotherapy modulation, but it has important methodological, structural, and interpretative limitations that should be addressed before publication. ​

Overall scope and positioning

The review successfully integrates data across gastrointestinal (GI) and extra‑GI malignancies, emphasizing common mechanisms (inflammation, genotoxicity, barrier dysfunction, immune modulation) and translational themes such as biomarkers, diet, probiotics, and faecal transplantation (FMT). However, it is essentially a narrative review despite using quasi‑systematic language (“Data Extraction and Synthesis”) without a transparent search strategy, inclusion/exclusion criteria, or risk‑of‑bias assessment, which weakens its evidentiary weight and may overstate causal inferences.

Methodology and literature handling

  • The “Data Extraction and Synthesis” section is very generic and does not specify databases searched, time frames, search terms, study types included, or how duplicate and low‑quality studies were handled; the article reads as an expert narrative rather than a systematic or even scoping review and should be labelled and structured accordingly.
  • Multiple statements are framed causally (“microbiome-driven cancer”, “prostate exemplifies how systemic dysbiosis contributes to extra‑GI cancer risk”) based largely on associative or preclinical data; the authors rarely distinguish correlation, Mendelian randomisation, mouse models, and human interventional evidence, which risks overinterpretation for clinicians.
  • There is no explicit critical appraisal of key landmark trials (e.g., FMT–ICI melanoma studies, antibiotic–ICI interactions), nor discussion of conflicting data or negative trials, which gives the impression of a one‑sided, “optimistic” narrative about microbiota modulation. ​

Structure, coherence, and redundancy

  • The GI vs extra‑GI division is clinically intuitive, and the per‑tumour subsections (oncogenesis vs therapy response) are consistent, facilitating reading.
  • There is frequent repetition of the same mechanistic motifs (Akkermansia, Faecalibacterium, F. nucleatum, bile acids, SCFAs, FMT) across sections with minimal synthesis, which could be condensed into a more integrative mechanistic chapter plus shorter, cancer‑specific vignettes. ​
  • Minor inconsistencies exist (the abstract and introduction mention 13 cancers, while the discussion states “ten cancers” and Table 1 is labelled as covering “ten cancers”), suggesting that sections were edited without fully harmonising the manuscript.

Depth, balance, and mechanistic interpretation

  • Mechanistic summaries are generally accurate but sometimes simplified to the point of being schematic (e.g., assigning clear “protective” vs “pro‑tumour” labels to taxa without acknowledging strain heterogeneity, context dependence, and diet/host genotype interactions). ​
  • Some areas are underdeveloped given current evidence:
    • Breast cancer: the role of estrobolome and breast tissue microbiota is mentioned but not contrasted with large human cohorts that show modest effect sizes and high inter‑individual variability. ​An example of a paper of local breast microbiota to cite: “Vilhais G, et al. Case report: Primary CDK4/6 inhibitor and endocrine therapy in locally advanced breast cancer and its effect on gut and intratumoral microbiota. Front Oncol. 2024 Mar 27;14:1360737. doi: 10.3389/fonc.2024.1360737. PMID: 38601755; PMCID: PMC11004348.”
    • Lung cancer: Mendelian randomisation data are highlighted but not critically discussed in terms of instrument strength, pleiotropy, or how far one can go from MR to clinical translation. ​
    • Brain tumours: the section extrapolates from general gut–brain axis literature and early glioma work; the text would benefit from more caution on the current absence of robust interventional microbiota data in neuro‑oncology.
  • The review underplays potential confounding by cancer stage, prior antibiotics, diet, and concomitant medications in interpreting microbiota associations, which is crucial for clinicians considering microbiota profiling. ​

Clinical and translational relevance

  • The clinical translation section is conceptually strong (biomarkers, diet, probiotics, FMT, personalised modulation) but remains high‑level and does not clearly distinguish:
    • What is already guideline‑relevant (e.g., avoiding unnecessary antibiotics around ICIs). ​
    • What is plausible but still experimental (FMT to rescue ICI response, engineered probiotics in trials). ​
    • What is speculative (precision “microbiota‑informed” systemic therapy selection).
  • Safety, regulatory issues, and standardisation challenges for FMT and live biotherapeutics are mentioned only superficially, despite being central barriers to clinical adoption. ​
  • The review does not provide practical guidance for oncologists (e.g., how to counsel patients on commercial probiotics, dietary fibre, or over‑the‑counter microbiota tests), missing an opportunity to translate the science into bedside considerations.

Figures, tables, and referencing

  • Figures appear conceptually useful but schematic; their added value over the text is modest, and some are partially redundant (e.g., Figures 4 and 5 both summarizing shared pathways).
  • Table 1 is mentioned as a “comparative summary” but is not visible in detail in the excerpt; however, the text suggests it is descriptive rather than critical (no indication of evidence strength, human vs preclinical data, or level of clinical readiness).
  • Referencing is extensive and mostly current (up to 2025), which is a strength, but key controversies (discordant findings across microbiota sequencing pipelines, geography‑dependent taxa differences, failure to replicate some “signature” taxa) are not discussed, reducing critical depth. ​

Style, terminology, and minor issues

  • The manuscript is generally well written and accessible, but at times uses promotional language (“perhaps the most exciting therapeutic avenue”, “benchmark cancer”) that is more appropriate for a commentary than a balanced review.
  • Terms such as “oncobiosis” and “microbiome‑driven cancer” are used without precise definition or caveats, which may suggest a stronger causal role than is currently justified.
  • There are minor typographical issues (e.g., “oncotherapies”, inconsistent spacing, “thera‑peutic”) and some lingering copy‑editing artifacts (placeholder DOI, “Academic Editor: Firstname Last‑name”) that need correction before acceptance.

Specific recommendations for improvement

  • Clarify the review type: explicitly label this as a narrative review and soften the quasi‑systematic phrasing in the methodology or alternatively provide a concise but real methods section (databases, time frame, keywords, inclusion/exclusion criteria).
  • Introduce a standardised “evidence strength” framework (e.g., preclinical only, early clinical association, interventional human data) across cancer‑specific sections and in Table 1, clearly differentiating hypothesis‑generating data from practice‑changing evidence. ​
  • Add a dedicated subsection on limitations and controversies: inter‑study heterogeneity, sequencing/platform issues, reverse causality, host factors (HLA, diet, BMI, medications), and negative/neutral trials of probiotics or dietary modulation in immunotherapy.
  • Rebalance tone by replacing causal or categorical statements (“microbiome-driven malignancy”, “exemplifies how dysbiosis contributes to risk”) with more cautious phrasing (“associated with”, “may contribute to”, “supported mainly by preclinical models”) where appropriate.
  • Enhance clinical relevance by summarising practical implications for oncologists:
    • When to be cautious with antibiotics around ICI;
    • Current stance on routine microbiota profiling.
    • Reasonable, evidence‑aligned counselling on diet and probiotics in different tumour types.

If these issues are addressed—particularly methodological transparency, clarification of causal claims, and a more critical appraisal of the literature—the review would offer a useful, clinically relevant synthesis of the microbiota–cancer field rather than a predominantly descriptive catalogue of associations.

Comments on the Quality of English Language
  • The manuscript is generally well written and accessible, but at times uses promotional language (“perhaps the most exciting therapeutic avenue”, “benchmark cancer”) that is more appropriate for a commentary than a balanced review.
  • Terms such as “oncobiosis” and “microbiome‑driven cancer” are used without precise definition or caveats, which may suggest a stronger causal role than is currently justified.
  • There are minor typographical issues (e.g., “oncotherapies”, inconsistent spacing, “thera‑peutic”) and some lingering copy‑editing artifacts (placeholder DOI, “Academic Editor: Firstname Last‑name”) that need correction before acceptance.

Author Response

Response to Reviewer 3

We sincerely thank Reviewer 3 for the thorough and insightful critique of our manuscript. We have carefully reviewed all comments and have revised the manuscript to address the reviewer’s major concerns where feasible, while also clarifying the intended scope and limitations of this narrative review. Our point-by-point responses are provided below.

Comment 1: Review type and methodological transparency

The review is essentially narrative, despite using quasi-systematic language (“Data Extraction and Synthesis”) and lacking a transparent search strategy, inclusion/exclusion criteria, or risk-of-bias assessment.

Response:
We thank the reviewer for highlighting this critical issue. In response, we have explicitly reframed the manuscript as a narrative review. The section previously titled “Data Extraction and Synthesis” has been revised to “Literature Selection and Narrative Synthesis.” In this section, we now clarify the databases searched (PubMed and Web of Science), the publication timeframe (2010–2025), and the qualitative nature of evidence synthesis. We also explicitly state that no formal risk-of-bias assessment was performed, consistent with the review's narrative scope.

Comment 2: Overstatement of causality

Multiple statements are framed causally based largely on associative or preclinical data.

Response:
We appreciate this critical observation. We have revised the manuscript to use more cautious causal language throughout, replacing terms such as “driver” and “microbiome-driven” with “associated with,” “linked to,” or “may contribute to.” In addition, clarifying language has been added in selected sections to indicate when conclusions are supported primarily by preclinical or observational evidence.

Comment 3: Lack of explicit evidence stratification and discussion of conflicting data

The review does not clearly distinguish between preclinical, associative, and interventional evidence, nor does it discuss conflicting or negative studies.

Response:
We thank the reviewer for this thoughtful comment. Given the broad scope of the manuscript and its design as a narrative review covering multiple cancer types, our primary aim was to synthesize common mechanistic and translational themes rather than to formally stratify evidence strength or to provide exhaustive discussion of conflicting or negative studies for each cancer type. We acknowledge this as a limitation inherent to narrative reviews and have moderated interpretative language accordingly to avoid overstatement of evidence.

Comment 4: Redundancy, limited synthesis, and optimistic tone

There is repetition of mechanistic motifs across sections with limited synthesis, and the tone may appear overly optimistic.

Response:
We appreciate this perspective. Because the review spans both gastrointestinal and extra-gastrointestinal malignancies, some repetition of key mechanistic pathways was intentional, allowing each cancer-specific section to remain self-contained for readers with organ-focused interests. Nevertheless, we have revised language where appropriate to ensure a balanced tone and to avoid promotional phrasing, while maintaining clarity and accessibility.

Comment 5: Clinical and translational relevance

The clinical translation section remains high-level and does not clearly distinguish guideline-relevant, experimental, and speculative applications.

Response:
We agree that clearer distinctions between established clinical practice and emerging investigational strategies are important. Given the narrative and hypothesis-generating nature of this review, we intentionally avoided providing guideline-level recommendations or detailed clinical algorithms. We have therefore revised wording to emphasize that microbiome-targeted interventions such as probiotics, dietary modulation, and fecal microbiota transplantation remain largely investigational and should not yet be considered routine clinical practice.

We are grateful to Reviewer 3 for the constructive feedback, which helped us improve the clarity, framing, and balance of the manuscript. We believe these revisions more accurately reflect the narrative scope of the review while maintaining scientific rigor and clinical relevance.